Manuscript prepared for Atmos. Chem. Phys.
with version 2014/09/16 7.15 Copernicus papers of the LaTeX class copernicus.cls.
Date: 12 July 2018

# Simulated and observed horizontal inhomogeneities of optical thickness of Arctic stratus

Michael Schäfer[1,*], Katharina Loewe[2,*], André Ehrlich[1], Corinna Hoose[2], and Manfred Wendisch[1]

[1]Leipzig Institute for Meteorology, University of Leipzig, Leipzig, Germany
[2]Institute of Meteorology and Climate Research, Karlsruhe Institute of Technology, Karlsruhe, Germany
[*]Both authors contributed equally to this work.

*Correspondence to:* Michael Schäfer (michael.schaefer@uni-leipzig.de), Katharina Loewe (katharina.loewe@kit.edu)

**Abstract.** Two-dimensional (2D) horizontal fields of cloud optical thickness $\tau$ derived from airborne measurements of solar spectral radiance during the Vertical Distribution of Ice in Arctic Clouds (VERDI) campaign (carried out in Inuvik, Canada in April/May 2012) are compared with semi–idealized Large Eddy Simulations (LES) of Arctic stratus performed with the COnsortium for Small-Scale MOdeling (COSMO) atmospheric model. The input for the LES is obtained from collocated airborne dropsonde observations. Four consecutive days of a persistent Arctic stratus observed above the sea–ice free Beaufort Sea are selected for the comparison. Simulations are performed for spatial resolutions of 50 m (1.6 km × 1.6 km domain) and 100 m (6.4 km × 6.4 km domain). Macrophysical cloud properties such as cloud top altitude and vertical extent are well captured by COSMO. Cloud horizontal inhomogeneity quantified by the standard deviation and one-dimensional (1D) inhomogeneity parameters show that COSMO produces more homogeneous clouds by a factor of two (100 m spatial resolution) compared to the measurements. Those differences reduce for the spatial resolution of 50 m. However, for both spatial resolutions the directional structure of the cloud inhomogeneity is well represented by the model. Differences between the individual cases are mainly associated with the wind shear near cloud top and the vertical structure of the atmospheric boundary layer. A sensitivity study changing the wind velocity in COSMO by a vertically constant scaling factor shows that the directional small–scale cloud inhomogeneity structures can range from 250 m to 800 m and depend on the mean wind speed, if the simulated domain is large enough to capture also large–scale structures, which then influence the small–scale structures. For those cases a threshold wind velocity is identified, which determines when the cloud inhomogeneity stops increasing with increasing wind velocity.

# 1 Introduction

Arctic clouds are expected to be a major contributor to the so-called Arctic Amplification (Serreze and Barry, 2011; Wendisch et al., 2017) and, therefore, need to be represented adequately in model projections of the future Arctic climate (Vavrus, 2004). Especially, low-level Arctic stratus are of importance (Wendisch et al., 2013), because they occur quite frequently (around 40 %, Shupe et al., 2006, 2011), typically persist over several days or even weeks (Shupe et al., 2011), and on annual average, warm the Arctic surface (Shupe and Intrieri, 2004). The numerous physical and microphysical processes that determine the properties of Arctic stratus are complexly linked to each other (e.g., Curry et al., 1996) and still not understood in full detail (Morrison et al., 2012).

Dynamic factors (updrafts), which increase the actual supersaturation in the cloud beyond the equilibrium values for both liquid water and ice, and a steady supply of water vapor from above the cloud act to stabilize the Arctic stratus (Shupe et al., 2008). This facilitates the simultaneous existence of both phases (Korolev, 2007). While in updrafts liquid and ice crystals grow, the cloud top cooling induces downward vertical motion, where Wegener-Bergeron-Findeisen process may dominate. Therefore, small–scale structures can be important to understand the microphysical processes. Additionally, Arctic stratus shows microphysical inhomogeneities, which typically occur on horizontal and vertical scales below a few kilometers and even tens of meters (Chylek and Borel, 2004; Lawson et al., 2010). The small–scale cloud structures, which accompany cloud inhomogeneities, lead to three-dimensional (3D) radiative effects (Varnai and Marshak, 2001), which can be parameterized using inhomogeneity parameters (Iwabuchi and Hayasaka, 2002; Oreopoulos and Cahalan, 2005).

Unfortunately, the understanding of Arctic clouds is impeded by a paucity of comprehensive observations due to a lack of basic research infrastructure and the harsh Arctic environment (Intrieri et al., 2002; Shupe et al., 2011). Therefore, observation of small–scale cloud structures within the Arctic circle are sparse. Satellite observations are typically too coarse to resolve scales below 250 m and space–born passive remote sensing observations suffer from contrast problems over highly reflecting surfaces (snow and ice, Rossow and Schiffer, 1991). Ground–based remote sensing observations with radar and lidar typically point only in zenith direction and are not capable to provide the horizontal 2D–structure of clouds. Only along the wind direction the variability of clouds is resolved (Shiobara et al., 2003; Marchand et al., 2007). For example, using correlation analysis, Hinkelmann (2013) revealed significant differences between along–wind and cross–wind solar irradiance variability on small spatial scales in broken–cloud situations. In comparison, airborne spectral imaging observation of reflected solar radiation provide areal measurements with spatial resolution down to several meters (Schäfer et al., 2015). Bierwirth et al. (2013) used such airborne measurements of reflected solar spectral radiance to retrieve fields of cloud optical thickness $\tau$ of Arctic stratus and demonstrated their strong spatial variability. From similar measurements, Schäfer et al. (2017a) analyzed the directional variability of different cloud types including Arctic stratus. The few analyzed cases revealed that 1D-statistics are not sufficient to quantify the variability of horizontal clouds in-

homogeneities.

Likewise, treating small–scale inhomogeneities using reanalysis data and atmospheric models is difficult. Global reanalysis products have relatively coarse spatial resolutions (40 km and larger; Lindsay et al., 2014) and, therefore, do not resolve small–scale features. Furthermore, in numerical weather prediction and climate models, the representation of the temporal evolution of mixed-phase clouds is poor (Barrett et al., 2017a, b). Especially, areas of up- and downdrafts in Arctic stratus,

which are typically in the range of less than 1 km cannot be resolved but have to be parametrized (Field et al., 2004; Klein et al., 2009). To realistically simulate the spatial structure of these clouds, Large Eddy Simulations (LES) with a spatial resolution of 100 m or less and high vertical resolution ($\approx 20$ m within atmospheric boundary layer, ABL) are needed. Those LES can resolve the vertical motion of the turbulent eddies in the ABL and the cores of up- and downdrafts representing the inho-

mogeneities in the cloud top structure, which can be seen in the amount of liquid water at the cloud top. The size of the up- and downdraft cores may differ depending on the time of the year (Roesler et al., 2016).

Previous LES studies focus for instance on cloud-top entrainment (Mellado, 2017) and emphasize the behavior of changes in the spatial resolution on the liquid water path (Pedersen et al., 2016).

Kopec et al. (2016) discussed two main processes, the radiative cooling and wind shear. The radiative cooling sharpened the inversion, while wind shear at the top of the ABL causes the turbulence in the capping inversion and lead to dilution at the cloud top.

In general, LES are helpful to focus on a certain process and to investigate cloud formation, cloud evolution or the small–scale structures in an Arctic stratus under controlled conditions. The further

aim is to characterize horizontal small–scale cloud inhomogeneities in the size range of less than 1 km in simulations and measurements to better understand the radiative properties of Arctic mixed-phase clouds. Results from the COSMO (COnsortium for Small-Scale MOdeling) model, which is adjusted to a LES setup with a high horizontal and vertical resolution to resolve the cloud structures of Arctic stratus (Loewe et al., 2017; Stevens et al., 2017) are evaluated. For the Arctic Summer

Cloud Ocean Study (ASCOS), Loewe et al. (2017) validated COSMO for simulations with a spatial resolution of 100 m with respect to droplet/ice crystal number concentrations, cloud top/bottom boundaries, and surface fluxes. Cloud structures and inhomogeneities were not validated due to the lack of observational data. Here, airborne imaging spectrometer measurements obtained during the VERDI campaign are used to analyze the small–scale cloud inhomogeneities ($< 1$ km), which are

then compared to COSMO simulations using the same model setup as proposed by Loewe et al. (2017) with 64 by 64 grid points and 100 m spatial resolution as well as a finer resolved setup with 32 by 32 grid points and 50 m spatial resolution. For that, data measured by dropsondes served as input for semi–idealized simulations of clouds using COSMO-LES (Sec. 2.3 and Sec. 3). Airborne measured fields of cloud optical thickness retrieved from imaging spectrometer measurements

(Sec. 2.2) are used for a comparison with the resulting COSMO clouds with respect to their overall

cloud inhomogeneity and directional features of the cloud inhomogeneities (Sec. 4 and Sec. 5). Observations and modelling are aimed to be combined to quantify the horizontal cloud top structures, which are discussed in Sec. 5 and Sec. 6.

## 2 Airborne measurements

### 2.1 VERtical Distribution of Ice in Arctic clouds (VERDI) campaign

Cloud remote sensing and atmospheric profiles by dropsondes from the airborne VERDI campaign (Bierwirth et al., 2013; Schäfer et al., 2015, 2017a) conducted in April/May 2012 are exploited in this study. VERDI was based in Inuvik, Canada. All data were observed aboard the Polar 5 research aircraft of the Alfred–Wegener–Institute, Helmholtz Centre for Polar and Marine Research (AWI). The measurement flights were mainly carried out in the region over the Beaufort Sea, which was mostly covered by sea ice but also included sea-ice free areas (Polynias). Mostly stratiform low level liquid and mixed-phase clouds within a temperature range of -19°C to 0°C where investigated (Costa et al., 2017). Here, the analysis is focused on a persistent cloud layer probed on four consecutive days from 14 to 17 May 2012. The applied measurements were performed in close vicinity ($\leq 50\,\text{km}$) over constant surface conditions (open water; Polynias). The persistent cloud layer in the respective area decreased continuously from day to day with cloud top altitude decreasing from about 880 m on 14 May to around 200 m on 17 May (Klingebiel et al., 2015; Schäfer et al., 2015, 2017a).

The Polar 5 research aircraft was equipped with a set of cloud and aerosol in situ and remote sensing instruments (Bierwirth et al., 2013; Schäfer et al., 2015; Klingebiel et al., 2015). Atmospheric profiles of temperature, humidity, wind speed and direction were derived from dropsonde measurements, which were regularly released during all flights.

### 2.2 Horizontal fields of cloud optical thickness

The qualitative and quantitative description of the cloud inhomogeneities is performed using fields of cloud optical thickness $\tau$. Marshak et al. (1995), Oreopoulos et al. (2000), or Schröder (2004) proposed to study horizontal cloud inhomogeneities using cloud-top reflectances. However, Schäfer et al. (2017a) pointed out that radiance measurements include the information of the scattering phase function (e.g., forward–/backward scattering peak, halo features). To avoid artifacts in the inhomogeneity analysis from such features, parameters that are independent of the directional scattering of the cloud particles have to be analysed. Therefore, to characterize the observed and simulated cloud fields regarding their horizontal cloud inhomogeneities the cloud optical thickness is applied, which does not include the fingerprint of the scattering phase function.

The 2D fields of $\tau$ used for the comparison with COSMO are retrieved from 2D fields of reflected solar spectral radiance, which were collected with the imaging spectrometer AisaEAGLE (Schäfer et al., 2013, 2015). Using those data, Schäfer et al. (2017a) retrieved ten fields of cloud optical thick-

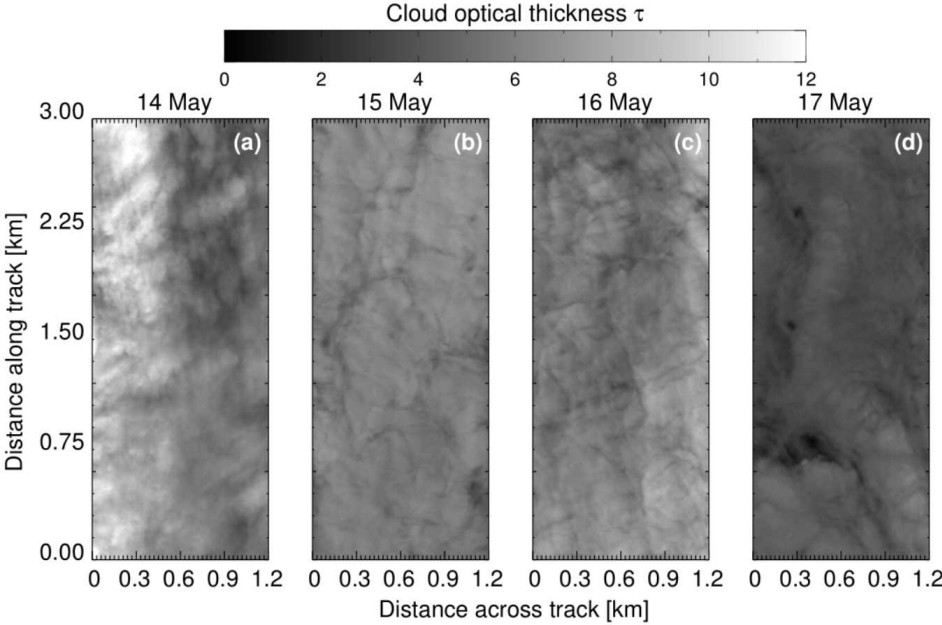

**Figure 1.** Exemplary selected sections (1.2 by 3.0 km) of horizontal fields of $\tau$ to illustrate the daily variability of the horizontal cloud inhomogeneities during the VERDI campaign on **(a)** 14 May 2012, **(b)** 15 May 2012, **(c)** 16 May 2012, and **(d)** 17 May 2012. Data adapted from Schäfer et al. (2017b).

ness $\tau$ (data set published on PANGAEA, Schäfer et al., 2017b). From those available ten fields of $\tau$, four cases are selected for the comparison to the LES results obtained from COSMO. Figure 1 exemplary illustrates selected sections (1.2 by 3.0 km) of the four chosen cases. The full widths and lengths of the applied fields of $\tau$ range to up to 1.7 km and 26.8 km, respectively. Their spatial resolution is 2.6 to 3.6 m (depending on the distance between aircraft and cloud).

During the time period from 14 to 17 May 2012, $\tau$ decreased from $8.1 \pm 1.2$ to $4.3 \pm 0.4$ (compare Tab. 2, Schäfer et al., 2017a). The selected sections in Fig. 1 illustrate the influence of the temporal evolution on the cloud features. In particular, from 15 to 17 May 2012 a reduction of the horizontal cloud inhomogeneity occurs, which is confirmed by Schäfer et al. (2017a). They also found a continuous reduction of cloud inhomogeneity during those four consecutive days. Furthermore, directional

features, which are prominent on 14 May, seem to be reduced, which is confirmed by autocorrelation analysis performed by Schäfer et al. (2017a).

## 2.3 Atmospheric profiles

During each measurement flight Vaisala dropsondes (type RD94) were used together with the Vaisala AVAPS (Airborne Vertical Atmosphere Profiling System) dropsonde receiving system (Hock and

Franklin, 1999; Coleman, 2003). The dropsondes were released to sample profiles of meteorological

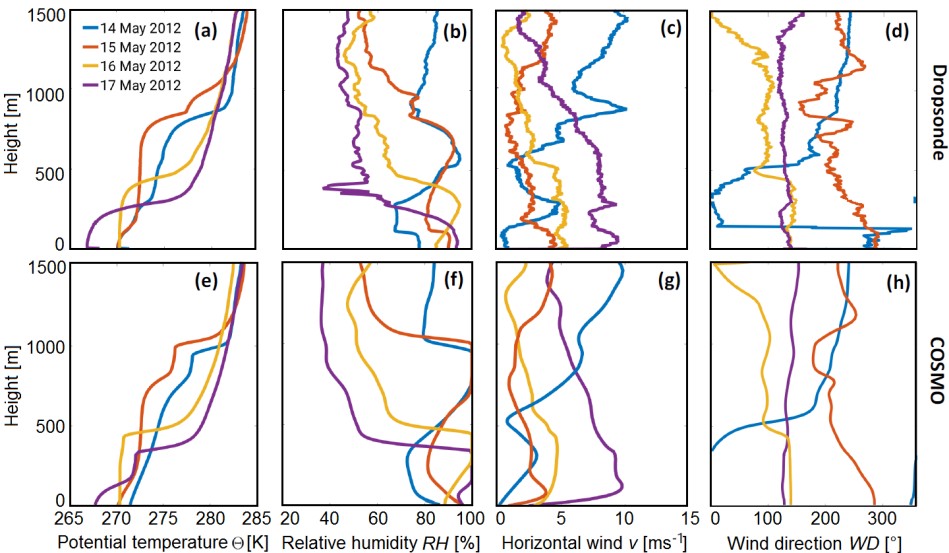

**Figure 2. (a, e)** Potential temperature, **(b, f)** relative humidity, **(c, g)** wind speed, and **(d, h)** wind direction for the four investigated cases. The dropsonde data is shown in the first row **(a-d)** and the 2 h domain-averaged profiles after spin-up time of the simulations are shown in the second row **(e-h)**. Dropsondes were released closest to the imaging spectrometer measurements.

parameters (air pressure $p$, air temperature $T$, relative humidity $RH$, wind speed $v$, and wind direction WD) below the aircraft, which then was typically operating at about 3 km altitude and allowed to sample the entire cloud and ABL structure by the dropsondes. The accuracy of the dropsonde measurements is given by the manufacturer and specified to $\pm\,0.4$ hPa for the air pressure, $\pm\,0.2°$C for the air temperature, $\pm\,2\,\%$ for the relative humidity, and $\pm\,0.5$ m s$^{-1}$ for the detected wind speed. For the analysis of the cloud fields, the dropsonde releases closest to the four investigated remote sensing observations had been chosen. The potential temperature ($\Theta$), relative humidity ($RH$), wind speed ($v$), and wind direction ($WD$) profiles for the four investigated cases are displayed in Fig. 2. From 14 May to 15 May the cloud top inversion increased from 810 m to 880 m while for the subsequent two days, the inversion layer decreased to 440 m on 16 May and to 200 m on 17 May 2012. In conjunction with the decrease of the cloud top altitude the cloud base altitude decreased as well until it almost reached the surface on 17 May. The relative humidity, displayed in Fig. 2b confirms the initial increase and consecutive decrease of the cloud top and base altitude. The inversion strength increased over the time period from $\approx 5$ K to $\approx 1$ K mainly because the temperature of the surface layer continuously decreased; the ABL became more stable.

Furthermore, Fig. 2c illustrates that the near-surface wind increased during the four days from $\approx 1$ to $\approx 10$ m s$^{-1}$, which might be of interest in terms of the generation of cloud inhomogeneities. Except for the case on 14 May, where wind speeds in higher altitudes are larger compared to the other days,

the daily increase of the near-surface wind speed is also observed in higher altitudes to up to 1 km.
Following Jacobson et al. (2013), this is related to Low–Level–Jets (LLJ) for the days from 15 to
17 May.

## 3 Simulations

### 3.1 COSMO: General setup

COSMO is a non-hydrostatic, limited-area atmospheric forecast model (Schättler et al., 2015). Here
it is used in a semi-idealized LES setup, which follows the description by Loewe et al. (2017), based
on Ovchinnikov et al. (2014) and Paukert and Hoose (2014). The two-moment cloud microphysics
scheme by Seifert and Beheng (2006) predicts the number densities and the masses of six hydrome-
teor types. The different ice phase hydrometeor growth processes are parameterized in this scheme.
In COSMO, the radiative transfer is described by a two-stream radiation scheme after Ritter and Ge-
leyn (1992). It is calculated every 2 s and has a direct cloud–radiative feedback. A three-dimensional
prognostic turbulence scheme describes the turbulent fluxes of heat, momentum and mass by a first–
order closure after Smagorinsky and Lilly (Herzog et al., 2002; Langhans et al., 2012). The size of
the model domain used by Loewe et al. (2017) was $6.4 \times 6.4$ km in horizontal direction with a spatial
resolution of 100 m. Here, this setup is applied as well. However, analyzing cloud inhomogeneities
requires a fine horizontal spatial resolution of the model simulations. Therefore, for the comparison
with the imaging spectrometer measurements analyzed here, the spatial resolution is also increased
to 50 m for addition model runs. In those cases, the domain size is reduced to 32 by 32 grid points
($1.6$ km $\times$ $1.6$ km) for computational constrains. A further reduction of the spatial resolution was not
possible due to numerical instabilities. The vertical height range of 22 km is divided into 166 verti-
cal levels, which are more dense for the ABL with a typical vertical resolution of around 15 m up
to the inversion height of the different days of investigation. The initialization profiles of tempera-
ture, humidity, wind speed, and wind direction are based on the dropsonde data. The dropsonde data
are partly affected by horizontal variability, when slowly passing the cloud and drifting horizontally.
Therefore, parts of the original profiles (Fig. 2) are smoothed and brought to a vertical monotonically
increasing profile for initialization of the model. The surface of the model is sea water and the sur-
face fluxes depend on the surface temperature, which is 273.5 K for the sea–water surface. Moreover,
ERA (European Reanalysis) -Interim reanalysis data (from the European Centre for Medium-Range
Weather forecast (ECMWF)) (Dee et al., 2011) have been used to complete the profiles above the
altitude where the dropsondes were released. Other model parameters such as the description of the
large scale subsidence, which is adjusted to the temperature inversion height, the relaxation to fixed
cloud droplet number concentration (CDNC) and ice crystal number concentration (ICNC), and the
spin up time of 2 h follows Ovchinnikov et al. (2014). The CDNCs are based on measurements of
the Small Ice Detector mark 3 (SID3) measurements (Vochezer et al., 2016). During the four inves-

**Table 1.** Model setup specifications of the different mixed-phase cloud simulations of four VERDI campaign days.

| Case | $z(T_{\mathrm{in}})$ [m] | CDNC [cm$^{-3}$] | ICNC [l$^{-1}$] |
|---|---|---|---|
| 14 May | 870 | 100 | 1 |
| 15 May | 988 | 100 | 1 |
| 16 May | 440 | 90 | 1 |
| 17 May | 350 | 100 | 1 |

tigated days, CDNC of 90 to 100 cm$^{-3}$ were observed as summarized in Tab. 1. Unfortunately, the
concentration of ice crystals was below or at the detection limit of the SID3. Therefore, the ICNC
were assumed to be one particle per liter according to observations of mixed–phase Arctic stratus
during the Indirect and Semi-Direct Aerosol Campaign (ISDAC) (McFarquhar et al., 2011; Ovchin-
nikov et al., 2014). The inversion height of the temperature $z(T_{\mathrm{in}})$ is necessary for the description of
the large–scale subsidence in the model and is represented by the inversion height of the dropsonde
profiles, which are used for initialization of the model simulations (Tab. 1).

### 3.2 Domain-averaged cloud properties and temporal evolution

Time series of simulated liquid water content (LWC) and ice water content (IWC) for the four se-
lected cases are shown in Fig. 3. During the four flights, which are simulated with COSMO, only few
ice crystals were observed. In terms of the model domain average profiles of the LWC and IWC, the
simulated clouds consist mostly of liquid water droplets except for the 15 May, in which more IWC
is built from around 4 h on (Fig. 3b). Furthermore, the cloud top is around 1000 m for the 14 May and
the 15 May (Fig. 3a, b). However, the cloud top height increases during time in all four simulations
because of entrainment of air through the top of the ABL. This is evident in the temporal evolution
of LWC, which has a maximum between 0.25 and 0.35 g kg$^{-1}$ near the cloud top. The Arctic clouds
on 16 May and 17 May are the lowest simulated clouds with a cloud top initially around 450 m and
350 m, respectively (Fig. 3 c, d).

The four simulations show differences in the temperature, relative humidity and wind speed profiles
(Fig. 2e–g), which in general still agree with the initial dropsonde profiles after the spin up time
(Fig. 2a–c). The height of the ABLs and the strength of the inversions are lower in the simulations of
the 16 May and 17 May. Furthermore, for the simulation on 17 May a second inversion develops in
the ABL near the surface around 60 m to 150 m. The ABL structure is well mixed in the simulation
of the 16 May although no second temperature inversion is built near the surface. The simulation of
the 16 May shows a wind shear from around 150° to around 100° (Fig. 2g) and a decrease of $v$ with
height above the cloud top height, which is also seen in the dropsonde profiles (Fig. 2c). The other
simulations do not show a turning of the wind directly above the inversion height.

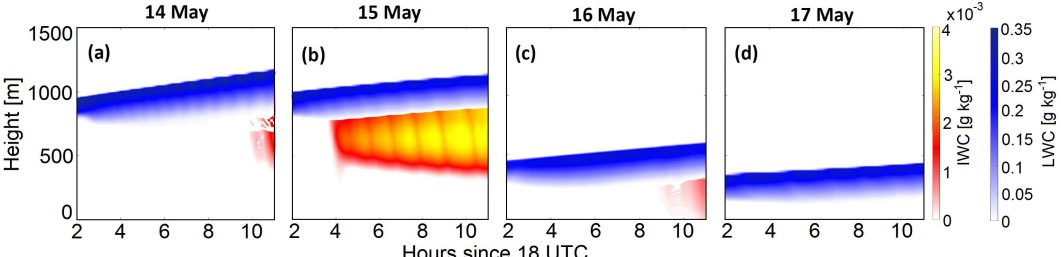

**Figure 3.** Domain averages of LWC (blue color scale) and IWC (red-yellow color scale) of the four simulations during the VERDI campaign. Please note the different color scale for the IWC in (d).

The simulated mixed-phase clouds of the four VERDI flights show a liquid water path (LWP) around 35 to $50\,\mathrm{g\,m^{-2}}$. The highest LWP is seen in the simulation of the 14 May, which increases towards $50\,\mathrm{g\,m^{-2}}$ at the end of the simulation. The simulation of the 15 May has the lowest LWP values. Furthermore, the LWP remains very stable until the end of the simulation. The ice water path (IWP) and the snow water path (SWP) of all four simulations is small especially for the simulated clouds on the 14, 16, and 17 May, which fits well with observations.

For the comparison of the simulated and observed horizontal cloud structures (cloud inhomogeneities), fields of simulated cloud optical thickness ($\tau_{\mathrm{sim}}$) are compared to retrieved fields of cloud optical thickness from the measurements ($\tau_{\mathrm{meas}}$). The $\tau_{\mathrm{sim}}$ is calculated within the COSMO model considering the amount of liquid water and the solar spectrum. However, it cannot be expected that COSMO is capable of reproducing the detailed spatial and temporal cloud evolutions, which are captured by the observed fields of $\tau$, accurately (inhomogeneity features and directional structures). Therefore, besides the comparison of observed and simulated clouds with regard to macrophysical cloud features (cloud vertical extent, cloud optical thickness) of the individual cases, instead of point-by-point comparisons of cloud parameters, statistical bulk parameters describing the horizontal cloud inhomogeneities, their directional structures, and the temporal evolution of both will be compared.

## 4 Quantification of cloud inhomogeneities

### 4.1 One-dimensional statistical bulk parameters

For the quantitative description of the cloud inhomogeneities from the simulated fields of cloud optical thickness ($\tau_{\mathrm{sim}}$) obtained from COSMO and measurement-based retrieved fields of cloud optical thickness ($\tau_{\mathrm{meas}}$) collected during the VERDI campaign, statistical techniques are applied. Following Schäfer et al. (2017a), different statistical quantitative measures of the cloud inhomogeneities are derived using the mean and standard deviation of the particular $\tau$ field and three 1D inhomogeneity parameters $\rho_\tau$ (Davis et al., 1999b; Szczap et al., 2000), $S_\tau$ (Davis et al., 1999b; Szczap et al., 2000),

and $\chi_\tau$ (Cahalan , 1994; Oreopoulos and Cahalan, 2005). They are given by:

$$\rho_\tau = \frac{\sigma_\tau}{\bar{\tau}}, \tag{1}$$

$$S_\tau = \frac{\sqrt{\ln\left(\rho_\tau^2 + 1\right)}}{\ln 10}, \tag{2}$$

$$\chi_\tau = \frac{\exp\left(\overline{\ln\tau}\right)}{\bar{\tau}}. \tag{3}$$

A homogeneous cloud is characterized by $\rho_\tau = 0$ and $S_\tau = 0$. Higher values of $\rho_\tau$ and $S_\tau$ indicate more pronounced cloud inhomogeneity. However, both of them have no predefined upper limit. Therefore, $\rho_\tau$ and $S_\tau$ only sustain a quantitatively significance, when their values for different cases

are compared to each other. The 1D inhomogeneity parameter $\chi_\tau$ ranges between 0 and 1, with values close to unity indicating horizontal homogeneity and values approaching zero characterizing high horizontal inhomogeneity. Due to the limited range between 0 and 1, $\chi_\tau$ is not only a qualitative but also quantitative measure.

### 4.2 Two-dimensional autocorrelation analysis

Two-dimensional autocorrelation analysis is applied to quantify the typical scales of cloud inhomogeneities and to identify directional patterns of the cloud structure (Schäfer et al., 2017a). To derive the autocorrelation functions, each field of $\tau$ is correlated with itself, while it is shifted pixel by pixel (observations) or grid point by grid point (simulations) against itself. The values of the resulting correlation coefficients after each shift are in the range between -1 (perfect negative correlation)

and 1 (perfect positive correlation). Correlation coefficients with values of 0 identify no correlation. Here, only the degree of correlation matters, not if it has a positive or negative sign. Similar to Schäfer et al. (2017a), squared autocorrelation functions $P_\tau^2$ are used to avoid ambiguous interpretations. The $P_\tau^2$ reach values between 0 (no correlation) and 1 (perfect correlation).

The particular correlation coefficients at the derived distances identify the similarity of the horizontal

cloud structures. If the cloud is horizontally homogeneous, the correlation coefficients stay constant over large distances. If the cloud is rather inhomogeneous the correlation coefficients already drop at closer distances. Therefore, $P_\tau^2$ as a function of distances is a measure of the size of the dominant cloud structures.

A quantitative value for the distance at which cloud structures are different from each other (namely

decorrelated) is the decorrelation length $\xi_\tau$ (Schäfer et al., 2017a). It is the distance at which $P_\tau^2$ drops to:

$$P_\tau^2(\xi_\tau) = \frac{1}{e^2}. \tag{4}$$

In a 2D-autocorrelation function, $\xi_\tau$ can differ depending on the orientation, if the cloud structures have a predominant orientation. To quantify this directionality, $\xi_\tau$ is calculated along ($\xi_\tau^{\updownarrow}$) and across ($\xi_\tau^{\leftrightarrow}$) the predominant direction. The larger the differences between $\xi_\tau^{\updownarrow}$ and $\xi_\tau^{\leftrightarrow}$, the more cloud structures are orientated.

Figure 4a shows a section of an observed field of $\tau_{\mathrm{meas}}$, retrieved from the measurements on 15 May. The selected section has a swath of 1.3 km (oriented in y direction) and a length of 6 km (oriented in x direction). Figure 4b shows the corresponding field of $\tau_{\mathrm{sim}}$ (6 km $\times$ 6 km, adapted to the selected length of the measurement case), which is simulated with COSMO two hours after the spin up time for the case on 15 May. For comparability reasons, both fields of $\tau$ are normalized by their maximum.

Although the swath (y direction) of the field of $\tau_{\mathrm{meas}}$ is smaller by a factor of almost five compared to the field of $\tau_{\mathrm{sim}}$, larger cloud structures of similar size and shape are obvious in both fields of $\tau_{\mathrm{meas}}$ and $\tau_{\mathrm{sim}}$. However, with 488 spatial pixels along the swath (spatial double binning was applied during measurements) and a field of view of 37° AisaEAGLE's spatial resolution is $\approx 1.3$ m for a target in a distance of 1 km. Thus, the spatial resolution of AisaEAGLE is relatively high, compared to the spatial resolution of 100 m from COSMO. Thereby, the exact pixel size of AisaEAGLE depends on the distance between aircraft and cloud, which leads to pixel sizes between 2.6 and 3.6 m for the four investigated cases. Due to the 30 to 40 times higher spatial resolution of AisaEAGLE, compared to COSMO, the measurements shows cloud features, which cannot be resolved by COSMO. Those features on a spatial scale below 100 m may have an effect on the statistical (1D inhomogeneity parameters) and spatial comparison (autocorrelation analysis) of the particular fields of $\tau$.

To quantify the size and orientation of the represented cloud structures in the observations and simulations, Fig. 4c and Fig. 4d show the calculated squared 2D autocorrelation coefficients $P_\tau^2$. To calculate them, different numbers of legs (shifts) have to be applied for $P_{\tau,\mathrm{meas}}^2$ and $P_{\tau,\mathrm{sim}}^2$. The applied field of $\tau_{\mathrm{meas}}$ consists of $2700 \times 450$ spatial pixels. Therefore, restricted to the shorter side, $225 \times 225$ (half of swath pixel number, calculated into x and y direction) legs are chosen for the calculation of the 2D $P_{\tau,\mathrm{meas}}^2$. COSMO consists of $64 \times 64$ grid points. This allows $32 \times 32$ legs for the calculation of $P_{\tau,\mathrm{sim}}^2$.

The resolved domain and spatial resolution displayed in Fig. 4c and Fig. 4d show significant differences, which reveals that a direct comparison is difficult. Applying the 2D autocorrelation analysis to the observations allows to resolve small-scale cloud structures with high spatial resolution ($\approx 2.7$ m), but only within a narrow spatial range below 1 km. Contrarily, the same analysis for COSMO delivers $P_{\tau,\mathrm{sim}}^2$ with lower spatial resolution ($\geq 100$ m), but over a larger spatial range ($\leq 3.2$ km, in Fig. 4d only displayed until 2 km). Thus, also large–scale cloud structures are covered by COSMO (purple stripes in Fig. 4d) but not in the observations. Therefore, the large–scale structures cannot be compared between observations and simulations. With respect to a comparison of the small–scale structures, the spatial sizes (spatial resolution, domain size) of both datasets need to be conformed

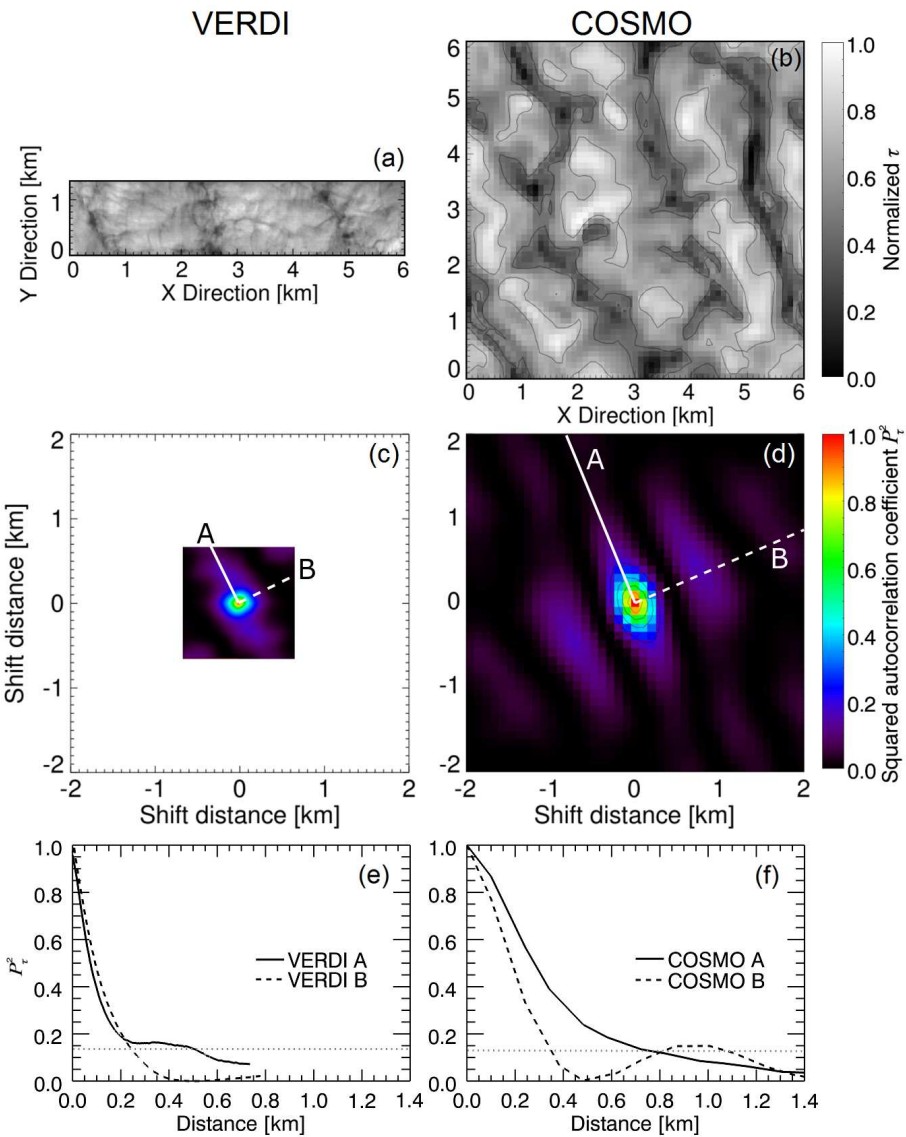

**Figure 4. (a–b)** Horizontal fields of normalized $\tau_{\mathrm{meas}}$ (VERDI) and $\tau_{\mathrm{sim}}$ (COSMO) for the case on 15 May 2012. **(c–d)** Two-dimensional autocorrelation coefficients $P^2_{\tau,\mathrm{meas}}$ and $P^2_{\tau,\mathrm{sim}}$, calculated for fields of $\tau$ displayed in (a) and (b). **(e–f)** One-dimensional autocorrelation coefficients along (straight white line marked in (c) and (d)) and across (dashed white line marked in (c) and (d)) predominant directional structure. The grey dotted line illustrates the threshold for the estimation of $\xi^{\updownarrow}_{\tau}$ and $\xi^{\leftrightarrow}_{\tau}$.

to make a direct comparison possible.

Furthermore, both, Fig. 4c and Fig. 4d show predominant directional features of the cloud structures. Their lengths and widths are derived from 1D autocorrelation functions along (straight white line in Fig. 4c and Fig. 4d) and across (dashed white line in Fig. 4c and Fig. 4d) those predominant directional structures and a subsequent estimation of $\xi^{\updownarrow}_{\tau}$ and $\xi^{\leftrightarrow}_{\tau}$. The derived $\xi^{\updownarrow}_{\tau}$ and $\xi^{\leftrightarrow}_{\tau}$ show an overall

agreement but still differ from each other. For the observations $\xi_{\tau,\text{meas}}^{\updownarrow}$ and $\xi_{\tau,\text{meas}}^{\leftrightarrow}$ reach distances of $\approx 500\,\text{m}$ and $\approx 250\,\text{m}$, respectively. Contrarily, for the simulations $\xi_{\tau,\text{sim}}^{\updownarrow}$ and $\xi_{\tau,\text{sim}}^{\leftrightarrow}$ reach distances of $\approx 800\,\text{m}$ and $\approx 400\,\text{m}$, respectively. This is a further indication that it is necessary to make the fields of $\tau_{\text{meas}}$ and $\tau_{\text{sim}}$ conform with respect to their spatial resolution and domain. In the following this is done by (i) averaging the observed fields of $\tau_{\text{meas}}$ to the spatial resolution of the simulated

fields of $\tau_{\text{sim}}$ and (ii) improving the spatial resolution of the simulations itself.

    Figure 4e and Fig. 4f further illustrate that it is not possible to compare the large–scale structures between observations and simulations. The large–scale structures, which are covered by the COSMO simulations are identified by a second increase of the $P_{\tau,\text{sim}}^2$ at distances ($\approx 1\,\text{km}$ in Fig. 4f) larger than $\xi_\tau$. The width of the measured fields is too narrow to cover such a second increase in the $P_{\tau,\text{meas}}^2$

(compare Fig. 4e). Therefore, the further comparison of the cloud structures, which are identified in the observations and simulations, is restricted to the small–scale cloud structures with sizes below 1 km only.

### 4.3   Final data preparation - Adjustment of spatial resolution and domain

    To compare both data sets, the fields of $\tau_{\text{meas}}$, which are retrieved from the imaging spectrometer

measurements are averaged to the spatial resolution of the COSMO $\tau_{\text{sim}}$ fields. The investigations on the single cases during VERDI are performed for spatial resolutions of 50 m (32 by 32 grid points) and 100 m (64 by 64 grid points). All other model parameters are kept constant with respect to the analysis performed by Loewe et al. (2017).

    In order to average the observed fields of $\tau_{\text{meas}}$ to the spatial resolution of 50 and 100 m, the $\tau_{\text{meas}}-$

values of distinct numbers of neighboring pixels are averaged. The number depends on the single pixel size of the particular cases, which is a function of the distance between aircraft and cloud. For the four investigated cases this number varies between 13 (26) and 18 (36) pixels, which are needed to generate pixel sizes of $\tau_{\text{meas}}$ comparable to the 50 m (100 m) spatial resolution of COSMO.

    Furthermore, for the simulations with 100 m spatial resolution, the domain size of the measurements

and simulations need to be adapted. The applied COSMO domain size of 6.4 km by 6.4 km is about three to four times larger than the domain size of the measurements. Therefore, to compare both data sets, the COSMO domain size is also reduced to the width and length of the corresponding $\tau_{\text{meas}}$ field from the measurements. Therefore, for the comparison, only a squared domain in the center of COSMO's $\tau_{\text{sim}}$ field is used, which size corresponds to the size of the particular field from

the measurement. For the four investigated cases this results in COSMO domains composed out of $12 \times 12$ to $16 \times 16$ grid points ($1.2 \times 1.2\,\text{km}$ to $1.6 \times 1.6\,\text{km}$). Longer stripes of $\tau_{\text{meas}}-$fields and stripes according to their lengths across the COSMO domain are not used, because the investigations are focused on small scale cloud inhomogeneities, which are already covered by the smaller squared domain size given by the swath of the $\tau_{\text{meas}}-$fields.

For the COSMO simulations, which use 50 m spatial resolution, the domain size is reduced to 32 by

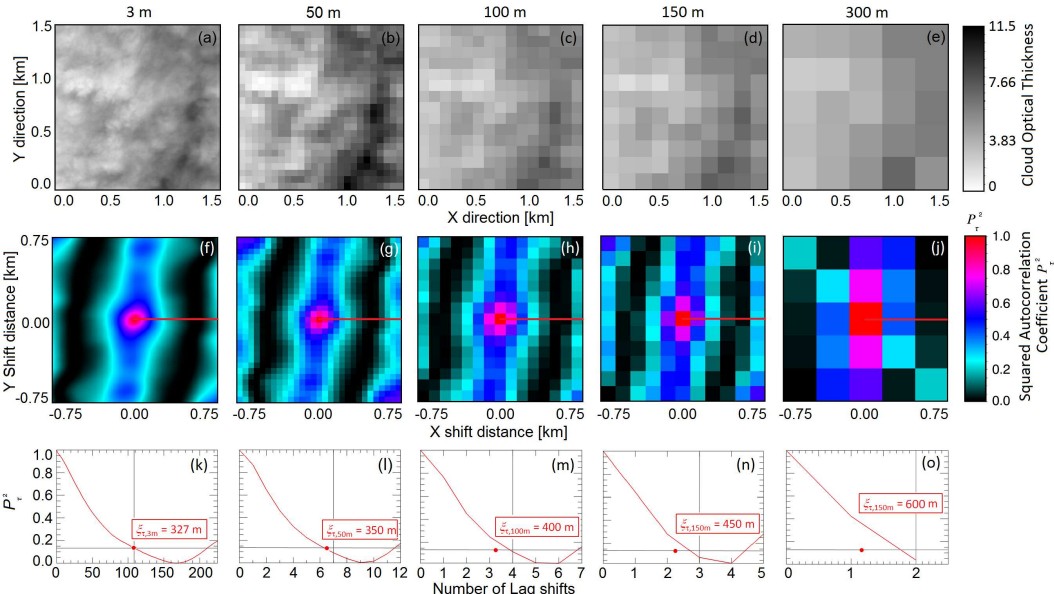

**Figure 5.** Illustrated are sections of one and the same field of $\tau_{\mathrm{meas}}$ from 14 May 2012 with a spatial resolutions of **(a)** $\approx 3\,\mathrm{m}$ (original resolution), **(b)** $50\,\mathrm{m}$ (COSMO resolution), **(c)** $100\,\mathrm{m}$ (COSMO resolution), **(d)** $150\,\mathrm{m}$, and **(e)** $300\,\mathrm{m}$. **(f–j)** Squared 2D autocorrelation coefficients $P_\tau^2$ calculated for the fields of $\tau_{\mathrm{meas}}$ displayed in (a) to (e). **(k–o)** Squared 1D autocorrelation coefficients $P_\tau^2$ calculated along straight red line in (f) to (j). Estimated decorrelation length $\xi_\tau$ is marked by horizontal and vertical black line and labeled by its value. Red dot marks $\xi_\tau$ as derived from the case with the original spatial resolution of $3\,\mathrm{m}$.

32 grid points resulting in a total domain of $1.6\,\mathrm{km}$ by $1.6\,\mathrm{km}$, which is comparable to the observations. Therefore, the domain of those simulations was not adapted for the comparisons.

However, to increase the statistics, which might be otherwise too small because of the finally applied small domain but large pixel sizes, for COSMO averages of the resulting $P_{\tau,\mathrm{sim}}^2$ over all output time steps after spin up are used. For the measured fields, which lengths are much longer than their widths, squared domains (size determined by swath of $\tau_{\mathrm{meas}}$) are cut along the measured stripe and the resulting $P_{\tau,\mathrm{meas}}^2$ are averaged accordingly. Increasing the number of available $P_{\tau,\mathrm{meas}}^2$ to average is a further restriction to use squared domains instead of stripes.

To test possible effects arising from the change of spatial resolution and to check if the relevant scales of cloud inhomogeneity are lost, when reducing the resolution of the measurements, Fig. 5a to Fig. 5e show sections of one and the same field of $\tau_{\mathrm{meas}}$ from 14 May, but displayed with a different spatial resolution of $3\,\mathrm{m}$ (original resolution), $50\,\mathrm{m}$ (COSMO fine resolution), $100\,\mathrm{m}$ (COSMO original resolution), $150\,\mathrm{m}$, and $300\,\mathrm{m}$ resolution. Figure 5f to Fig. 5j show the corresponding squared 2D autocorrelation coefficients. The red line illustrates the direction, which is used to calculate the squared 1D autocorrelation functions and decorrelation lengths $\xi_\tau$ displayed in Fig. 5k to Fig. 5o. The fields from the 2D autocorrelation analysis show that except for the spatial resolution of $300\,\mathrm{m}$

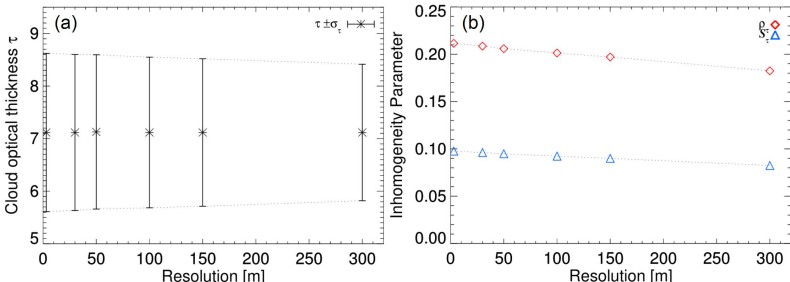

**Figure 6.** Comparison of **(a)** mean and standard deviation and **(b)** inhomogeneity parameters $\rho$ and $S$ as a function of spatial resolution for the fields of $\tau_{\mathrm{meas}}$ illustrated in Fig. 5a–e.

the directional structure of the cloud inhomogeneities is still captured, when the spatial resolution is reduced. However, the decorrelation lengths, derived from the 1D autocorrelation analysis increases with decreasing spatial resolution from $\xi_\tau = 327\,\mathrm{m}$ at $3\,\mathrm{m}$ spatial resolution to $\xi_\tau = 600\,\mathrm{m}$ at $300\,\mathrm{m}$

spatial resolution. Therefore, decreasing spatial resolution leads to larger $\xi_\tau$, which indicates larger cloud structures. This means that reduced spatial resolution will generate fields of $\tau$ with larger spatial scales.

To test the influence of the spatial resolution on the overall inhomogeneity, Fig. 6a shows the results for the mean and standard deviation of the fields of $\tau$, illustrated in Fig. 5. Figure 6b shows the cor-

responding 1D inhomogeneity parameters $\rho_\tau$ and $S_\tau$. While the mean value of $\tau$ stays constant for all spatial resolutions, its standard deviation decreases with increasing pixel size. This indicates that the fields of $\tau$ become more homogeneous the larger the pixel size is. Similarly, the value of both 1D inhomogeneity parameters $\rho_\tau$ and $S_\tau$ decrease with increasing pixel size.

Therefore, in the following analysis, comparing the simulated against observed fields of $\tau$, the sim-

ulations with the finer spatial resolution of $50\,\mathrm{m}$ are used. The simulations with $100\,\mathrm{m}$ spatial resolution are used to discuss the model sensitivity with respect the spatial resolutions.

## 5   Comparison of modeled against observed cloud structures

### 5.1   Magnitude of inhomogeneity

The fields of $\tau$ obtained from the spectral imaging remote sensing ($\tau_{\mathrm{meas}}$) are compared to the

fields of $\tau$ derived from the COSMO simulations ($\tau_{\mathrm{sim}}$). To validate the cloud inhomogeneity in the simulated fields, the statistical techniques from Sect. 4.1 including the averaging of the measured fields to 50 and $100\,\mathrm{m}$ pixel size are applied. Table 2 lists the mean value of $\tau$, standard deviation $\sigma_\tau$, and the three 1D inhomogeneity parameters $\rho_\tau$, $S_\tau$, and $\chi_\tau$ for the observations and the simulations with the two different spatial resolutions of 50 and $100\,\mathrm{m}$.

Both, measurements and simulation show the highest cloud optical thickness on 14 May with

**Table 2.** Mean value of $\tau$, standard deviation $\sigma_\tau$, and the three 1D inhomogeneity parameters $\rho_\tau$, $S_\tau$, and $\chi_\tau$ calculated for all four cases from the observations and the simulations with the two different spatial resolutions of 50 and 100 m.

| | Case | $\bar{\tau} \pm \sigma_\tau$ | $\rho_\tau$ | $S_\tau$ | $\chi_\tau$ |
|---|---|---|---|---|---|
| VERDI (50 m) | 14 May | $7.8 \pm 1.5$ | 0.195 | 0.086 | 0.979 |
| | 15 May | $6.4 \pm 0.7$ | 0.121 | 0.055 | 0.992 |
| | 16 May | $6.4 \pm 1.0$ | 0.166 | 0.078 | 0.983 |
| | 17 May | $4.2 \pm 0.5$ | 0.154 | 0.071 | 0.986 |
| VERDI (100 m) | 14 May | $8.1 \pm 1.2$ | 0.209 | 0.093 | 0.977 |
| | 15 May | $6.4 \pm 0.5$ | 0.115 | 0.052 | 0.993 |
| | 16 May | $6.6 \pm 0.6$ | 0.145 | 0.065 | 0.988 |
| | 17 May | $4.3 \pm 0.4$ | 0.132 | 0.061 | 0.990 |
| COSMO (50 m) | 14 May | $7.9 \pm 0.6$ | 0.071 | 0.030 | 0.997 |
| | 15 May | $7.1 \pm 0.7$ | 0.092 | 0.040 | 0.995 |
| | 16 May | $6.0 \pm 0.6$ | 0.094 | 0.040 | 0.995 |
| | 17 May | $5.8 \pm 0.5$ | 0.083 | 0.036 | 0.996 |
| COSMO (100 m) | 14 May | $6.9 \pm 0.5$ | 0.066 | 0.028 | 0.997 |
| | 15 May | $5.4 \pm 0.3$ | 0.053 | 0.023 | 0.998 |
| | 16 May | $5.5 \pm 0.5$ | 0.090 | 0.037 | 0.996 |
| | 17 May | $5.6 \pm 0.3$ | 0.044 | 0.019 | 0.999 |

$\bar{\tau}_{\mathrm{meas}} = 8.1 \pm 1.2$ and $\bar{\tau}_{\mathrm{sim}} = 7.9 \pm 0.6$ at 50 m spatial resolution and $\bar{\tau}_{\mathrm{sim}} = 6.9 \pm 0.5$ at 100 m spatial resolution, which show an overall agreement. During the course of the following days, the large scale subsidence lead to a decrease of the cloud top altitude and cloud geometrical thickness and corresponding lower values of $\tau$ and $\sigma_\tau$. For these days, model and observations are still in

agreement. However, compared to the spatial resolution of 100 m it is obvious that the finer resolved simulations lead to better agreements between measurements and simulations.

Regarding the cloud inhomogeneity, the absolute values of the 1D inhomogeneity parameters $\rho_\tau$, $S_\tau$, and $\chi_\tau$ do not compare well for the simulations with 100 m spatial resolution. The results for the COSMO simulations show lower 1D inhomogeneity parameters (more homogeneous) by a factor

of two and higher, compared to the results from the measurements. The agreement between the observations and simulations increase with the finer spatial resolution of 50 m, but still does not match perfectly. The reason might be that the comparably lower inhomogeneity derived from COSMO for both spatial resolutions is caused by its effective spatial resolution, which is approximately three times 50 m or accordingly three times 100 m (Skamarock et al., 2004). Although the pixel

size of AisaEAGLE is adapted to the COSMO spatial resolution by averaging over neighboring pixels, COSMO's effective spatial resolution is larger, which might lead to larger homogeneity of

the simulations compared to the observations. Furthermore, COSMO simulates the cloud at the same location, where it is initialized. Contrarily, the AisaEAGLE measurements took place along a stripe of several kilometers. The simulated clouds may not change in between the time steps as much as the measurements of the clouds along the measurement stripe do. Therefore, averaging over COSMO's time steps might further produce more homogeneous results than averaging over AisaEAGLE's squared domains along the flight track.

However, the observations show that the cloud field became more homogeneous from 14 to 15 May as indicated by lower values of $\rho_\tau$, which reduce from 0.209 to 0.115. From 15 to 16 May, $\rho_\tau$ increases to 0.145, which indicates a cloud field with slightly higher inhomogeneity. Then, on 17 May, $\rho_\tau$ reduced to 0.132, showing that the cloud field became more homogeneous again. These different cases with high and low $\rho_\tau$ are reproduced by COSMO independent on the chosen spatial resolution. Larger discrepancy between modeled and observed inhomogeneity parameters only occurred on 14 May, when the observations were influenced by large–scale cloud structures.

Nevertheless, the lower/higher inhomogeneity is also imprinted in the inhomogeneity parameters $S_\tau$ and $\chi_\tau$, which are smaller/larger in both, measurements and simulations, indicating that COSMO performs well with regard to the 1D inhomogeneity parameters.

## 5.2 Spatial inhomogeneity scale

The 2D autocorrelation functions are calculated to compare the typical spatial scales and the directional character of the small–scale cloud inhomogeneities (no large–scale inhomogeneities like roll convection) of observations and simulations. The 2D autocorrelation coefficients ($P^2_{\tau,\mathrm{meas}}$; $P^2_{\tau,\mathrm{sim}}$) for each case are shown in Fig. 7e to Fig. 7h for the measurements and in Fig. 7m to Fig. 7p for the simulations (50 m spatial resolution). Additionally, representative fields of normalized $\tau_{\mathrm{meas}}$ (Fig. 7a–d) and $\tau_{\mathrm{sim}}$ (Fig. 7i–l) are added. The 2D autocorrelation analysis was applied to the simulated fields of $\tau_{\mathrm{sim}}$ orientated in a North-South and West-East grid. The orientation of the observations is determined by the flight direction. Therefore, the orientation of the fields of $\tau_{\mathrm{meas}}$ and $P^2_{\tau,\mathrm{meas}}$ are rotated into the direction of the COSMO grid. One-dimensional $P^2_\tau$ are calculated manually along the dominant direction (straight red and blue lines in Fig. 7e–h and Fig. 7m–p) and across (dashed red and blue lines in Fig. 7e–h and Fig. 7m–p) it. For $P^2_{\tau,\mathrm{meas}}$ (red) and $P^2_{\tau,\mathrm{sim}}$ (blue) the results are displayed in Fig. 7i to Fig. 7l. The dotted black line illustrates the threshold for the estimation of $\xi_\tau$.

The observations on 14 May are influenced by a large scale cloud structure, which is caused by large scale dynamic forcing and leads to an increase of the autocorrelation coefficients for distances larger than 800 m. Furthermore, during this day a significant directional structure from North–West to South–East is observed. Along this direction the cloud field stays homogeneous over a wide range ($\xi_\tau = 800$ m). Across this predominant structure the small-scale cloud structures reach a decorrelation length of $\xi_\tau = 300$ m. During the following days the orientation of the directional structure turns

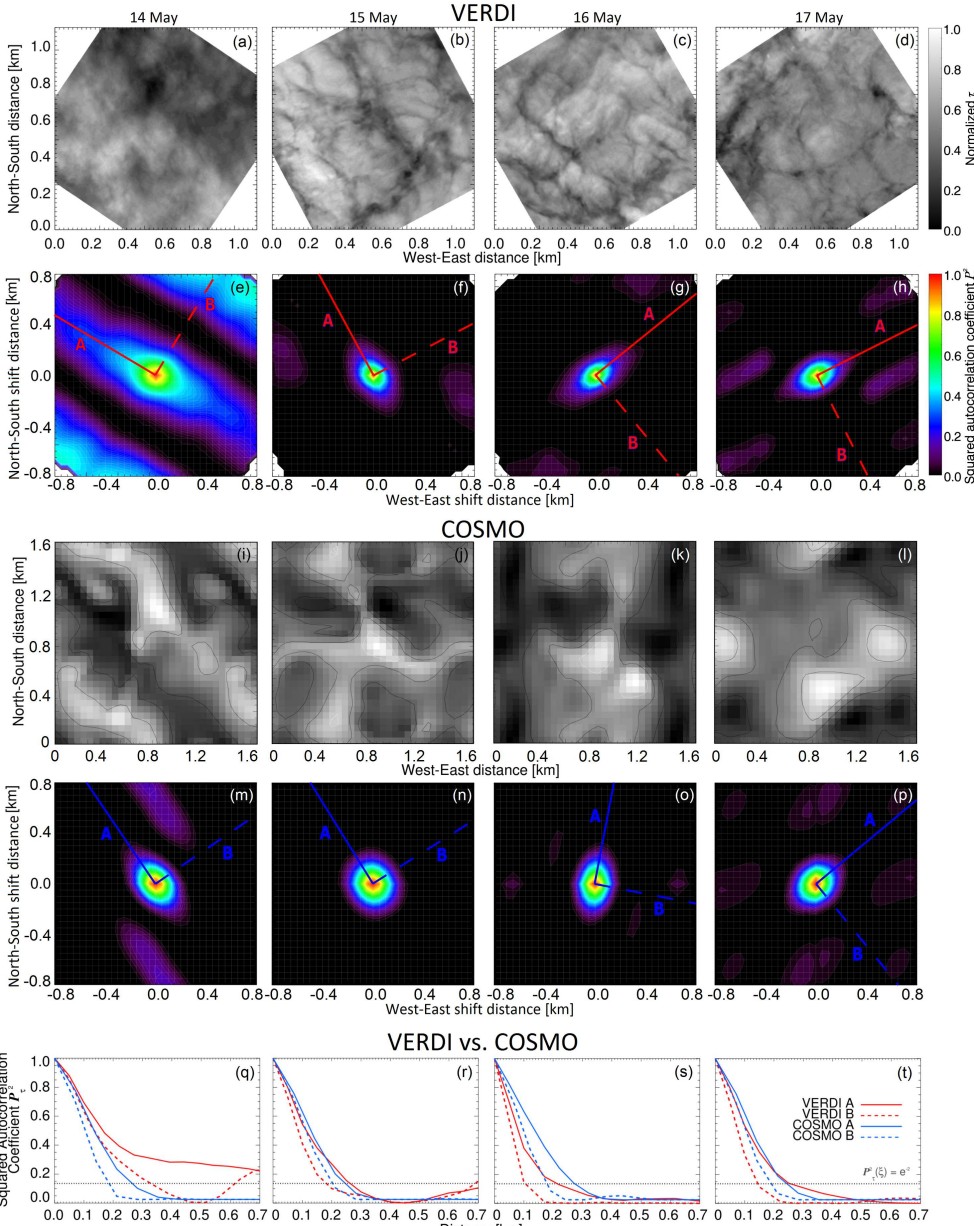

**Figure 7. (a-d)** Exemplary selected sections of fields of $\tau_{\mathrm{meas}}$ observed during VERDI from 14 to 17 May 2012. **(e–h)** Mean 2D autocorrelation coefficients $P^2_{\tau,\mathrm{meas}}$ derived for fields of $\tau_{\mathrm{meas}}$ from VERDI. **(i-l)** Exemplary selected fields of $\tau_{\mathrm{sim}}$ simulated with COSMO (50 m spatial resolution) for the VERDI cases from 14 to 17 May 2012. **(m–p)** Mean 2D autocorrelation coefficients $P^2_{\tau,\mathrm{sim}}$ derived for fields of $\tau_{\mathrm{sim}}$. **(q-t)** Decorrelation length $\xi_\tau$ along strongest (straight blue and red lines) and weakest (dashed blue and red lines) extend of 2D autocorrelation coefficients derived from $P^2_{\tau,\mathrm{meas}}$ in **(e–h)** and $P^2_{\tau,\mathrm{sim}}$ in **(m–p)**, respectively.

eastwards in the observations and the differences between $\xi_\tau^\updownarrow$ and $\xi_\tau^\leftrightarrow$ decrease. This characterizes a weakening of the directional structure of the cloud field.

Comparing the results for $P_{\tau,\mathrm{sim}}^2$ with $P_{\tau,\mathrm{meas}}^2$ reveals that the large scale cloud structure is not well simulated for the case on 14 May. This results most probably from the small domain size of COSMO, which is fixed over the same location when averaging the $P_{\tau,\mathrm{sim}}^2$ over a set of time steps. Contrarily, the averages of $P_{\tau,\mathrm{meas}}^2$ from the measurements are performed over a set of squared domains along the flight track. Thus, the chance to cover also larger structures is higher for the measurements compared to the simulations. However, the overall small–scale directional structures are well simulated. On 14 May, a significant directional structure from North–West to South–East is observed, which then turns eastwards for 15 to 17 May. Except on 16 May, the predominant simulated directions of the cloud fields are almost identically to the observations.

Furthermore, the results for $P_{\tau,\mathrm{meas}}^2$ and $P_{\tau,\mathrm{sim}}^2$ show that COSMO simulations using a spatial resolution of 50 m produce similar sizes of the small-scale cloud structures compared to the measurements. In Fig. 7m to Fig. 7p the covered areas of $P_{\tau,\mathrm{sim}}^2$ are of similar sizes compared to the areas covered by $P_{\tau,\mathrm{meas}}^2$ in Fig. 7e to Fig. 7h. Table 3 lists the resulting $\xi_{\tau,\mathrm{meas}}$ and $\xi_{\tau,\mathrm{sim}}$ calculated along ($\xi_\tau^\updownarrow$) and across ($\xi_\tau^\leftrightarrow$) the predominant structures found in Fig. 7e–h and Fig. 7m–p. A comparison reveals only minor differences between $\xi_{\tau,\mathrm{meas}}$ and $\xi_{\tau,\mathrm{sim}}$. The best agreement is achieved on 15 and 17 May, when $\xi_{\tau,\mathrm{meas}}$ and $\xi_{\tau,\mathrm{sim}}$ show almost identically results. On 16 May the differences are slightly larger, while on 14 May the differences are significantly larger, which might result from the insufficient simulated large–scale cloud structure. For the simulations with 100 m spatial resolution (graph not shown) the directional features still compare well between observations and simulations. Like for the measurements on 14 May a predominant North–West to South–East direction is simulated, which then turns eastwards. Thereby, the cases on 14 May and 16 May show the strongest directional features (largest differences between $\xi_\tau^\updownarrow$ and $\xi_\tau^\leftrightarrow$, compare Tab. 3) with $\xi_\tau^\updownarrow$ on 14 May larger than the width of the observed field of $\tau_{\mathrm{meas}}$. Although on 17 May COSMO simulates a more isotropic structure ($\xi_\tau^\updownarrow \approx \xi_\tau^\leftrightarrow \approx 400\,\mathrm{m}$) of the cloud inhomogeneities compared to the measurements ($\xi_\tau^\updownarrow = 370\,\mathrm{m} \neq \xi_\tau^\leftrightarrow = 260\,\mathrm{m}$) it captures the reduction of the overall directionality. Therefore, the overall results with regard to the directional structure provided by COSMO are acceptable. However, the covered areas of the 2D autocorrelation functions, where the values of $P_{\tau,\mathrm{sim}}^2$ are higher than $e^{-2}$ are larger compared to the areas covered by the particular $P_{\tau,\mathrm{meas}}^2$. Therefore, the $\xi_{\tau,\mathrm{meas}}$ and $\xi_{\tau,\mathrm{sim}}$ calculated along ($\xi_\tau^\updownarrow$) and across ($\xi_\tau^\leftrightarrow$) the predominant structures do not compare well (compare Tab. 3). Like expected from Fig. 5, the values from the simulations (except for $\xi_\tau^\updownarrow$ on 14 May) are larger compared to the values from the observations by 20 to 30 %.

**Table 3.** Calculated decorrelation lengths $\xi_{\tau,\mathrm{meas}}$ and $\xi_{\tau,\mathrm{sim}}$ for the two different spatial resolutions of 50 and 100 m along ($\xi_\tau^{\updownarrow}$) and across ($\xi_\tau^{\leftrightarrow}$) the observed and simulated predominant directions (compare Fig. 7e–h and Fig. 7m–p for 50 m spatial resolution).

|  | Case | $\xi_{\tau,50\,\mathrm{m}}^{\updownarrow}$ [m] | $\xi_{\tau,50\,\mathrm{m}}^{\leftrightarrow}$ [m] | $\xi_{\tau,100\,\mathrm{m}}^{\updownarrow}$ [m] | $\xi_{\tau,100\,\mathrm{m}}^{\leftrightarrow}$ [m] |
|---|---|---|---|---|---|
| VERDI | 14 May | 800 | 330 | > 1000 | 400 |
|  | 15 May | 260 | 180 | 280 | 190 |
|  | 16 May | 220 | 100 | 350 | 170 |
|  | 17 May | 250 | 150 | 370 | 260 |
| COSMO | 14 May | 260 | 190 | 530 | 320 |
|  | 15 May | 250 | 200 | 380 | 260 |
|  | 16 May | 270 | 180 | 500 | 280 |
|  | 17 May | 240 | 190 | 430 | 390 |

## 6 Sensitivity Study

The reasons for the differences on 16 May (Fig. 7s) are most probably related to the wind field and the temperature profile. Figure 2d and Fig. 2h illustrate the temporally averaged wind directions in the simulations. While the wind direction does not changed at the cloud top of the 14, 15, and 17 May, the simulation of the 16 May shows a turning of the wind. Together with the well-mixed ABL (Fig. 2) this case shows a typical example of a cold air outbreak roll convection (e.g., Brümmer , 1999). On 16 May the simulated wind speed is significantly higher compared to the other days, resulting from the initial conditions in the dropsonde profile (Fig. 2c,d). Influences from the surface fluxes are only expected if the cloud is coupled to the surface and if so, affect only the LWP of the cloud (Loewe, 2017). For de–coupled clouds, it is assumed that the cloud structure depends more strongly on the wind shear, respectively the wind speed. However, since the wind speed, wind direction, and temperature profile are the only parameters, which have been changed in the model input, the wind speed and wind shear are expected to be main drivers for the degree of horizontal cloud inhomogeneity.

To test its influence on the horizontal cloud inhomogeneity, the simulations for 15 May (50 and 100 m spatial resolution) are repeated for different initializations, where the wind profile is varied. Here, the case on 15 May is chosen, because it shows the best agreement between observations and simulations (Fig. 7r) to serve as a benchmark case. Based on the original wind profile, the wind speeds at each altitude are multiplied by (a) 0.5, (b) 1.0, (c) 1.5, (d) 2.0, (e) 2.5, and (f) 3.0. This leads to mean wind speeds (vertically averaged over cloudy region) of (a) $\approx 0.7\,\mathrm{m\,s^{-1}}$, (b) $\approx 1.5\,\mathrm{m\,s^{-1}}$, c) $\approx 2.2\,\mathrm{m\,s^{-1}}$, (d) $\approx 3.0\,\mathrm{m\,s^{-1}}$, (e) $\approx 3.7\,\mathrm{m\,s^{-1}}$, and (f) $\approx 4.4\,\mathrm{m\,s^{-1}}$. The wind shear was kept constant throughout all simulations.

Figure 8a to Fig. 8f show the simulated 2D fields of $\tau_{\mathrm{sim}}$ for the simulations with the domain size

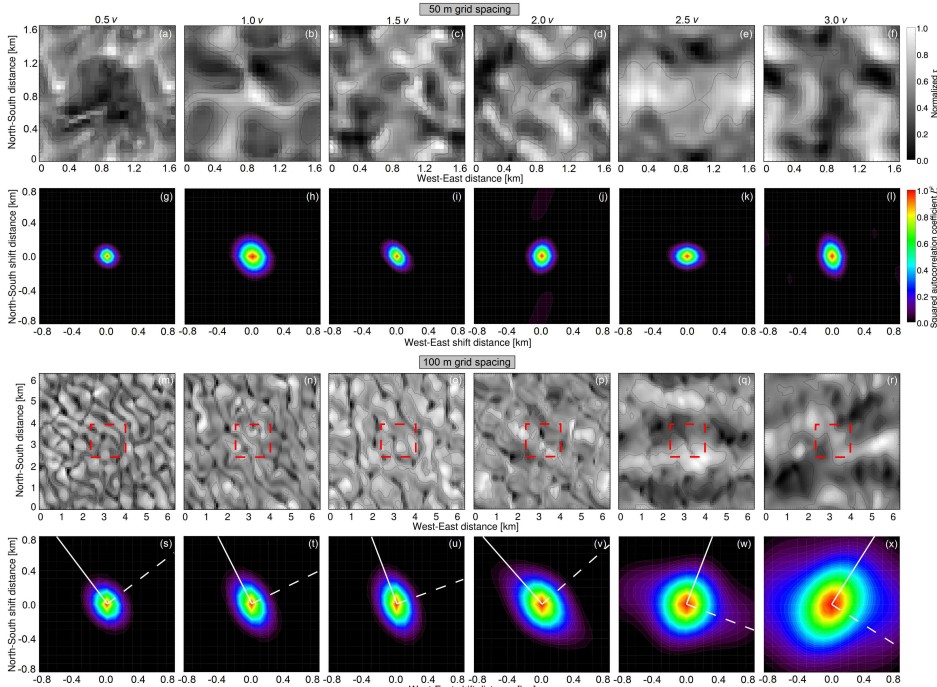

**Figure 8.** Exemplary selected fields of $\tau_{\mathrm{sim}}$ for the 15 May 2012 case simulated for differently scaled initial wind speeds on a grid with 50 m spatial resolution and 1.6 km by 1.6 km domain **(a–f)** and on a grid with 100 m spatial resolution and with 6.4 km by 6.4 km domain **(m–r)**. Calculated 2D autocorrelation coefficients $P_\tau^2$ are given for each case in **(g–l)** and **(s–x)**. White lines in (s)–(x) illustrate the orientation used for the calculation of the 1D $P_\tau^2$ along (straight white lines) and across (dashed white lines) the dominant directions illustrated in Fig.9. Red squares in (m)–(r) mark areas of comparable size to the small domains in (a)–(f).

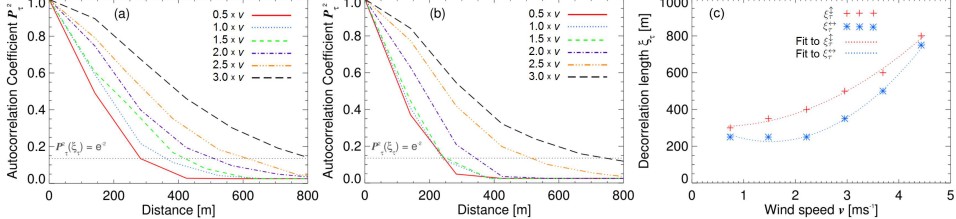

**Figure 9.** For the six cases of different wind speed calculated 1D autocorrelation functions **(a)** along and **(b)** across the main structures, identified in Fig. 8**(g)**–8**(l)**. The grey dotted line marks the threshold of $P_\tau^2(\xi_\tau) = \mathrm{e}^{-2}$. **(c)** From (a) and (b) derived discrete values for the decorrelation lengths $\xi_\tau^{\updownarrow}$ and $\xi_\tau^{\leftrightarrow}$ as a funktion of wind speed $v$ (symbols). Additionally included are fits derived from Eq. (5) and Eq. (6) (dotted lines).

of 1.6 km by 1.6 km and 50 m spatial resolution. Small–scale structures ($\leq 0.5$ km) are obvious and rather randomly orientated throughout the simulations for all six different initializing wind profiles. The spatial sizes of the small–scale structures quantified by the decorrelation length depend only

little on the wind speed. This is confirmed by the 2D autocorrelation analysis illustrated in Fig. 8g to Fig. 8l. Displayed are only the horizontal scales below 0.8 km, quantified by the 2D autocorrelation coefficients for shifts below $\pm 0.8$ km. A predominant direction of the small–scale structures is only slightly developed and varies independently from cases to case without clear preference. Furthermore, the $P_\tau^2$ and the decorrelation length, which vary between 150 and 300 m show only slight variations with changing wind speeds. This means that the sizes of the small–scale structures is basically independent to the wind speed.

Contrarily, the simulations with a domain size of 6.4 km by 6.4 km and 100 m spatial resolution show a clear dependency on the wind speed. The corresponding 2D fields of $\tau_{\rm sim}$ are illustrated in Fig. 8m to Fig. 8r. The small–scale structures ($\leq 0.5$ km) are still obvious in the simulations with coarse resolution, but for lower wind speeds, these small–scale structures have a North–West to South–East orientation, which turns into North–East to South–West orientation with increasing wind speeds. Additionally, large–scale structures ($\geq 2$ km), orientated perpendicular to the small–scale structures occur at $2.5 \times v$. The direction of these large-scale structures turns as well and becomes more obvious with increasing wind speeds.

The related results for the 2D autocorrelation analysis are given in Fig. 8g to Fig. 8l. With increasing wind speeds the area covered by $P_\tau^2 \geq P_\tau^2(\xi_\tau)$ increases. This illustrates that with increasing wind speed the size of the small–scale cloud structures increases along the predominant directions. The increased wind speed leads to stretched cloud structures along one direction. Along this predominant direction the stretching of the cloud structures smooths their variability stronger than across this direction. This leads to more homogeneous cloud structures. The turn of the orientation of the cloud structures to the East with increasing wind speed is also represented by the fields of $P_\tau^2$.

For the simulations with 100 m resolution, the dependency of the small–scale cloud structures on the wind speed was parameterized. Therefore, quantitative values for the size of the cloud inhomogeneity structures in terms of the decorrelation length $\xi_\tau$ and as a function of initialization wind speed are displayed in Fig. 9a (along predominant direction) and in Fig. 9b (across predominant structure). The threshold of $P_\tau^2(\xi_\tau) = e^{-2}$ is marked by a grey dotted line. The derived values for $\xi_\tau^{\updownarrow}$ and $\xi_\tau^{\leftrightarrow}$ are displayed in Fig. 9c as a function of the vertical mean wind speed within the cloudy region. It shows that along the predominant structure the decorrelation length $\xi_\tau^{\updownarrow}$ increases continuously (slightly quadratic increase) with increasing wind speed. Therefore, the derived decorrelation length along ($\xi_\tau^{\updownarrow}$) the predominant structure as a function of wind speed (vertically averaged over cloudy region) in units of m s$^{-1}$ can be approximated by:

$$\xi_\tau^{\updownarrow} = 31 \cdot v^2 - 31 \cdot v + 315. \tag{5}$$

Across the predominant structure (Fig. 9c) it is different, which means that for the lower wind speeds ($< 2 \times v$) no influence on $P_\tau^2$ and $\xi_\tau$ occurs, while it is comparable (slightly quadratic increase) to the values along the predominant structures for the stronger wind speeds ($\geq 2 \times v$). The derived decorrelation length across ($\xi_\tau^{\leftrightarrow}$) the predominant structure as a function of wind speed can

be approximated by:

$$\xi_\tau^\leftrightarrow = 60 \cdot v^2 - 183 \cdot v + 365. \tag{6}$$

Both, $\xi_\tau^\updownarrow$ and $\xi_\tau^\leftrightarrow$ characterize the small–scale cloud inhomogeneities. Large–scale cloud structures cannot be represented due to the too small domain size. However, comparing $\xi_\tau^\updownarrow$ with $\xi_\tau^\leftrightarrow$ shows that the directionality of the cloud structures first increases (0.5 to $2.0 \times v$) and afterwards decreases (2.0

to $3.0 \times v$) again. For the case investigated here, the threshold at $2.0 \times v$ applies to a mean $v$ (vertically averaged over cloudy region) of $3.0\,\mathrm{m\,s^{-1}}$.

Comparing the simulations for the small domain ($1.6 \times 1.6\,\mathrm{km}$, $50\,\mathrm{m}$ spatial resolution) with the large domain ($6.4 \times 6.4\,\mathrm{km}$, $100\,\mathrm{m}$ spatial resolution), indicates that the small–scale structures are most likely influenced by the large–scale structures. Only for the simulations with the large domain,

the small–scale structures depend on the wind speed. This indicates that small–scale cloud inhomogeneities are not directly linked to the wind speed but rather are influenced by the large–scale cloud inhomogeneities such as cloud roles. If these large–scale structures are not covered by the simulations (too small domain), the natural behavior of the small–scale structures (e.g. their size and orientation) might be disturbed. With respect to the comparison between observations and sim-

ulations, this may explain why only on 14 May larger differences between model and observations were found. All other three cases did not show a significant large–scale cloud structure, while on 14 May cloud roles were observed by the imaging spectrometer. Thus, the simulations of 15, 16, and 17 May are more uncritical with respect to the model domain than for 14 May, when a large domain is required to reproduce the large–scale cloud structures and, therefore, improve the simulation of the

small–scale cloud structures.

## 7   Summary and Conclusions

Cloud remote sensing of cloud optical thickness and atmospheric dropsonde measurements (profiles of air pressure, temperature, relative humidity, wind vector) from the airborne VERDI campaign conducted in April/May 2012 are exploited. In particular, a persistent cloud layer was analyzed, which

was probed on four consecutive days from 14 to 17 May 2012 in almost the same area ($\leq 50\,\mathrm{km}$) and over constant surface conditions (open water; Polynia). The cloud top altitude of the cloud layer shrank from day to day; it decreased from about $880\,\mathrm{m}$ on 14 May to around $200\,\mathrm{m}$ on 17 May. The airborne observations obtained during these days were applied to validate cloud simulations with COSMO by a new approach, which compares the observed and simulated 2D cloud fields.

The dropsonde profile measurements from the four consecutive days were used to initialize cloud simulations with COSMO. It is found that COSMO captures the measured cloud altitude, cloud vertical extent, and retrieved cloud optical thickness. The comparison of the horizontal small–scale cloud inhomogeneities identified within the observations and simulations was performed for horizontal fields of cloud optical thickness $\tau$. Those $\tau$ were either retrieved from airborne observations

of reflected solar radiances ($\tau_{\mathrm{meas}}$) or obtained from simulated 3D fields of LWC ($\tau_{\mathrm{sim}}$). For the reason of comparability, the observed fields of cloud optical thickness $\tau_{\mathrm{meas}}$ were aggregated to pixel sizes of 50 m and 100 m, the applied spatial resolutions of the individual simulations.

The general inhomogeneity was compared using 1D inhomogeneity parameters. For 100 m spatial resolution the absolute values of cloud inhomogeneity derived from COSMO are larger by a factor of about two, as compared to the values obtained from the observations. These differences slightly reduce, when the spatial resolution of the simulations is increased by a finer grid of 50 m. However, for both spatial resolutions the cloud inhomogeneity generated by COSMO is too low. This is mainly related to (i) the larger effective spatial resolution ($\approx 3 \times 50$ m and $\approx 3 \times 100$ m, respectively, Skamarock et al., 2004) of COSMO compared to the pixel size of the observations and (ii) a mismatch in timing/spacing, meaning that for the simulations by COSMO the 1D inhomogeneity parameters are averaged over several time steps over the same location, while for the observations the 1D inhomogeneity parameters are averaged over several time steps along the flight track. These results are in agreement with a model intercomparison by Ovchinnikov et al. (2014), who revealed that COSMO underestimates the variance of the vertical wind velocity compared to other LES models and, thus may cause an underestimation of the standard deviation of $\tau_{\mathrm{sim}}$. However, except for the case on 16 May the different magnitudes of cloud inhomogeneity of the individual days is well covered by COSMO.

Especially for the 14 May the cloud structure showed a distinct directional orientation, while from 15 to 17 May only a slight directional orientation is observed. Brümmer (1999) points out that such directed cloud structures are typical for Arctic stratus with cloud top altitudes below 1 km, which is the case here. Contrarily, for Arctic stratus with cloud top altitudes above 1.4 km cell structures are common. Based on a new method, proposed by Schäfer et al. (2017a), which is applied to COSMO data for the first time, a 2D analysis using autocorrelation functions is used to examine directional features of the cloud structures. The investigations showed that, in general, COSMO captured the observed directional structures of the cloud inhomogeneities. The wind directions of the individual cases showed a significant correlation to the direction of the predominant directional structures. During the four investigated days the orientation of the dominant directional structures within the observations turned eastwards by the same degree the wind direction changed. Similar results were found by (Houze, 1994), who stated that in case of changing wind shear cloud streets will be orientated along the mean wind direction.

The autocorrelation analysis was used to derive the characteristic size scale of the small–scale cloud structures by estimating the decorrelation length $\xi_\tau$, which is the distance at which the squared autocorrelation coefficients $P_\tau^2$ drop below $\mathrm{e}^{-2}$. The decorrelation lengths $\xi_\tau$ were calculated along ($\xi_\tau^{\updownarrow}$) and across ($\xi_\tau^{\leftrightarrow}$) the strongest extend of the derived $P_{\tau,\mathrm{meas}}^2$ and $P_{\tau,\mathrm{sim}}^2$. For the COSMO simulations with a spatial resolution of 50 m, the $\xi_\tau^{\updownarrow}$ and $\xi_\tau^{\leftrightarrow}$ agree well between observations and simulations, except for the case on 14 May. In contrast, for the simulations with a spatial resolution of 100 m,

COSMO produced small–scale cloud structures with characteristic sizes 20 to 30 % larger compared to the observations. However, for both spatial resolutions the best agreement was found for the case on 15 May 2012.

The agreement between COSMO and observations for the case on 15 May 2012 is used as basis for a systematic sensitivity study with respect to the wind speed as a main drivers of the cloud inhomogeneities. Simulations for the case on 15 May with differently scaled initialization wind profiles showed that the degree of horizontal cloud inhomogeneity was not significantly changed for the simulations with a small domain ($1.6 \, \text{km} \times 1.6 \, \text{km}$) and $50 \, \text{m}$ spatial resolution, but for the simula-

tions using a large domain ($6.4 \, \text{km} \times 6.4 \, \text{km}$) and $100 \, \text{m}$ spatial resolution. This indicates that the large–scale cloud structures such as cloud roles influence the small–scale cloud inhomogeneity. To correctly simulate the small–scale cloud inhomogeneity, COSMO needs to be run in a large domain, which also covers the large–scale cloud structures. This might have been the reason for the large differences between observations and simulations found for the case of 14 May, when pronounced

cloud rolls were observed. All other cases did not show such large–scale cloud structures and were simulated by COSMO closer to reality despite the small domain.

However, the significant impact of the wind on the small–scale cloud structures for simulations with $100 \, \text{m}$ spatial resolution confirms the importance of the wind speed for cloud inhomogeneities. For this case it was found that increasing wind speeds lead to larger horizontal cloud structures (in-

creased decorrelation lengths). A directionality of the cloud structures first increases ($0.5$ to $2.0 \times v$) and afterwards decreases ($2.0$ to $3.0 \times v$) with wind speed. A parameterization of the decorrelation lengths along and across the strongest autocorrelation with respect to the average wind speed in cloud altitude was derived, which can be used in future studies to generate cloud structures with specific sizes and shapes.

Furthermore, it is concluded that the wind direction and the atmospheric boundary layer structure are the explanation for the differences on 16 May. In contrast to the other three days a change of the wind direction of about $50°$ is found close to the cloud top. Additionally, the ABL was well mixed on 16 May, which increases the turbulent mixing within in the ABL and the cloud layer, and consequently influences the cloud top structure. Local differences in the wind fields at the position

where the dropsonde was released and the location where the imaging spectrometer measured might be the reason that this was not equally well captured by the simulations and measurements.

Altogether, cloud inhomogeneities are a challenge for cloud resolving models. Not only the spatially averaged magnitude of inhomogneity but also the directional structure and the interaction with large–scale cloud structures needs to be reproduced in the simulations. Although COSMO produces more

homogeneous clouds, it performed well, because it correctly represented the directional structures and the general degree of cloud inhomogeneity, if no larger–scale cloud structures are present. However, the statistical methods applied in this study can also be applied to characterize the larger–scale dynamic patterns, if the domain is large enough to resolve them.

## 8   Data availability

The fields of cloud optical thickness retrieved from the AisaEAGLE measurements are published on PANGAEA (Schäfer et al., 2017b). All other data used in this study are available upon request from the corresponding authors (michael.schaefer@uni-leipzig.de, katharina.loewe@kit.edu).

*Acknowledgements.* We thank Marco Paukert for introducing the COSMO setup. K. Loewe and C. Hoose acknowledge partial funding through the Helmholtz Programme "Atmosphere and Climate" and this project
has received funding from the European Research Council (ERC) under the European Union's Horizon 2020 research and innovation programme under grant agreement No 714062 (ERC Starting Grant "C2Phase"). We gratefully acknowledge the support by the SFB/TR 172 "ArctiC Amplification: Climate Relevant Atmospheric and SurfaCe Processes, and Feedback Mechanisms (AC)[3]" in Project B03 funded by the DFG. We thank Eike Bierwirth for organizing and performing the imaging spectrometer measurements during the VERDI campaign.
Furthermore, we thank Paul Vochezer, Martin Schnaiter and Emma Järvinen for providing the SID3 data. We are grateful to the Alfred Wegener Institute, Helmholtz Centre for Polar and Marine Research, Bremerhaven, Germany for supporting the VERDI campaign with the aircraft and manpower. In addition we like to thank Kenn Borek Air Ltd., Calgary, Canada for the great pilots who made the complicated measurements possible. For excellent ground support with offices and accommodations during the campaign we are grateful to the
Aurora Research Institute, Inuvik, Canada.

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
