# Peer review of "Simulated and observed horizontal inhomogeneities of optical thickness of Arctic stratus"

_Atmospheric Chemistry and Physics, 2018_

## Referee Comment (RC1) · Anonymous Referee #2 · 3 Apr 2018

General comments

In the manuscript the authors compare optical thickness of Arctic stratus cloud observed in the course of VERDI measurement campaign to that simulated by LES COSMO model. The authors conclude that the model produce clouds more homogeneous that the observed ones, yet the directional structures and the tendency of increasing /decreasing degree of inhomogeneity are reproduced in the simulations.

The attempt described in the manuscript is interesting, important and well described, however the results are not convincing. The real problem of the study is insufficient resolution of the numerical simulations to effectively match the experimental data. The majority of efforts is set to averaging of the experimental data which allows produce optical thickness fields of the resolution comparable to the model output due to limited

model domain and poor resolution of the simulations. This causes that conclusions are weak and too far going. Nevertheless, even weak and unconvincing results from very challenging efforts can be published on condition of critical analysis of the results and suggestions for improvements. I suggest a major revision of the paper before final acceptance.

Specific comments and suggestions for improvement

p. 5 Fig.2 Why you do not show wind components? Later you discuss directional shear...

3.1. Simulations Model set-up is not detailed enough. Please describe fluxes, radiation, microphysics in few sentences, referencing is not enough. Subversions on May 14-15 and 16-17 are at very different heights. Was vertical resolution at inversion height comparable?

p.6 l. 17 "to avoid numerical issues" really? Or data from dropsondes represent actual realization along trajectory, not a good choice for initial profiles?

p. 7 l. 12 WD in Fig 4 I guess is for wind direction, but generally the figure is hard to interpret. E.g. the same wind shear whether in the middle of the given colour and at the edge of colours can be visible or not. I fill not comfortable with this plot.

Section 4.2 The section shows nicely discrepancies between the experimental data and the simulation. Why in conclusion do you not call for higher resolution simulations? In the supplementary material of the paper you cite (Pedersen et al., 2016) there are suggestions that basic cloud patterns are reproduced reasonably in smaller domain. Why do you not perform sensitivity analysis due to model resolution?

Section 4.3 Again: your model domain larger than the swath of the measurements. Why not to run model in smaller domain but at higher resolution? In particular when you conclude that the decorrelation length increases with decreasing spatial resolution....

Section 5.1. I think that your conclusion that the motel captures temporal changes of

inhomogeneity is not well justified, there are only 4 points analysed. Moreover, the maximum modelled inhomogeneity is dated May 16th, while is observed on May 14th. On these days vertical profiles indicate that clouds ans boundary layer properties on these days are substantially different.

Section 5.2. Results in this section are more convincing. However, these results could be strengthened discussing dynamical patterns (convective rolls) th the boundary layer. Does the maximum optical thickness correlate with location of updraughts and maximum cloud top heights? Analysis of that could help to publish the paper, since conclusions are weak and should be supported with additional investigations which can increase our understanding of modelled processes. This is particularly important in terms of your sensitivity study in Section 6.

Section 6. Again: this section calls for more thorough analysis as pointed above.

Section 7. After the additional analyses this section (and abstract) should be updated adequately.

---

## Referee Comment (RC2) · Anonymous Referee #3 · 15 May 2018

Review of the article titled "Simulated and observed horizontal inhomogeneities of optical thickness of Arctic stratus" by Schafer and coauthors in the Atmospheric Chemistry and Physics.

The authors have used the data collected during the VERDI field campaign to derive optical thickness of mixed phased clouds. Then they have used the COSMO model to simulate the observed clouds with a goal of determining whether the model is able to simulate the observed inhomogeneity of cloud field and why. They conclude that the differences in the observed and simulated cloud fields are mostly due to differences in the wind speeds. The manuscript is well-written and will be of interest to the general meteorological community, and especially to those studying mixed-phase clouds. Below are some of my concerns regarding the manuscript.

Major Issues: The introduction is too long. Even after reading the second page, I am not sure why we should worry about the inhomogeneity in the cloud radiation properties aka optical depth. Is it because we need better sub-grid characterization of radiative properties in global climate models? It will be better if the authors explicitly state the specific objective of the study.

It is unclear why you chose wind speed as a tuning parameter. By increasing wind speeds you are simple changing the fluxes in the boundary layer and hence the turbulence. So essentially your results are suggesting that we greater turbulence produces higher inhomogeneity, which makes sense. It will be better if the authors can probe this. One way to tackle this would be to make some simulations where the winds are the same, but you increase the surface fluxes.

Lastly, the authors should show the comparison between the model reported liquid water paths, and cloud boundaries with those observed during the campaign. I think this will make the article complete. Thanks.

Minor Issues:

Line 58: Need reference to justify that sentence.

Line 101-102: I would simply say that the cloud fraction decreased. The word "dissolved" seems inappropriate in terms of clouds.

Line 121: By "ten fields" I believe you mean ten snapshots?

Section 3.1: Please describe the radiation and cloud schemes used in the model. Since you are evaluating optical depth, which is a radiative property, it is important to know this. Also mention how often the two schemes are talking to each other. Thanks.

Line 273-279: Please rephrase these sentences. It is confusing to read "large resolution" etc. thanks.

Figure 5e and 5f: There is no "grey dotted line" in the plot.

Figure 7a: the plot is showing mean and standard deviation, however there are two blue dots for each resolution? If you are showing mean+std and mean-std, then I suggest you show vertical error-bars.

[Figure]

---

## Author Comment (AC1) · 12 Jul 2018

We thank the reviewer for the helpful comments, which certainly improved the manuscript. Especially, by evaluating the different spatial resolutions of the observations and the simulations and associated effects we are sure that the manuscript became more meaningful. The detailed replies on the reviewer's comments are structured as follows. Reviewer comments have bold letters, are labeled, and listed always in the beginning of each answer followed by the author's comments including (if necessary) major revised parts of the manuscript. The revised parts of the manuscript are written in quotation marks and italic letters. Minor revisions of the text can be found in the additionally submitted mark-up file.

1. The real problem of the study is insufficient resolution of the numerical simulations to effectively match the experimental data. The majority of efforts is set to averaging of the experimental data, which allows produce optical thickness fields of the resolution comparable to the model output due to limited model domain and poor resolution of the simulations. This causes that conclusions are weak and too far going.

The reviewer is right that the differences between the original resolution of the AisaEAGLE observations and the COSMO simulations are a big issue. The primary reason for using the selected observations and the applied model resolution was based on a previous study by Loewe et al. (2017), where the COSMO model was compared to observations. For the 100 m grid spacing the comparisons of liquid/ice water content, size distributions of droplets and ice crystals showed good agreements. Therefore, we were confident with this model setup and first did not change the resolution for this study.

However, the reviewer is right that the large differences in the resolutions may not be comparable. Therefore, we added simulations, where we improved the grid spacing to 50 m. We tried to simulate with 25 m grid spacing, but this was not possible due to numerical instabilities. A reduction of the spatial domain could not solve this issue. A modification of the model, e.g. implementation of a different turbulence scheme, is beyond the scope of this work.

Throughout the discussion part of the manuscript, we now discuss the findings for both, 50 m and 100 m grid spacing. The comparison of both simulation runs did show significant differences, which show that the reproduction of small-scale cloud inhomogeneities depends on the model setup. In the revised manuscript this is explicitly analyzed and discussed. Including the new simulations with the new grid spacing of 50 m caused several changes in the manuscript. Please find below the main changes with regard to text parts, graphs, and tables:

**Abstract:**

"Simulations are performed for spatial resolutions of 50 m (1.6 km x 1.6 km domain) and 100 m (6.4 km x 6.4 km domain)."

"[...] show that COSMO produces more homogeneous clouds by a factor of two (100 m spatial resolution) compared to the measurements. Those differences reduce for the spatial resolution of 50 m."

**Introduction:**

"For the Arctic Summer Cloud Ocean Study (ASCOS), Loewe et al. (2017) validated COSMO for simulations with a spatial resolution of 100 m with respect to droplet/ice crystal number concentrations, cloud top/bottom boundaries, and surface fluxes. Cloud structures and inhomogeneities were not validated due to the lack of observational data. Here, airborne imaging spectrometer measurements obtained during the VERDI campaign are used to analyze the small–scale cloud inhomogeneities (

"Figure 5 (was Fig. 6). Illustrated are sections of one and the same field of  $\tau_{meas}$  from 14 May 2012 with a spatial resolutions of (a)  $\approx$  3 m (original resolution), (b) 50 m (COSMO resolution), (c) 100 m (COSMO resolution), (d) 150 m, and (e) 300 m. [...]"

**Section 5.1:**

|               | Case   | $\bar{\tau} \pm \sigma_{\tau}$ | $ ho_{	au}$ | $S_{\tau}$ | $\chi_{	au}$ |
|---------------|--------|--------------------------------|-------------|------------|--------------|
| VERDI (50 m)  | 14 May | $7.8\pm1.5$                    | 0.195       | 0.086      | 0.979        |
|               | 15 May | $6.4\pm0.7$                    | 0.121       | 0.055      | 0.992        |
|               | 16 May | $6.4\pm1.0$                    | 0.166       | 0.078      | 0.983        |
|               | 17 May | $4.2\pm0.5$                    | 0.154       | 0.071      | 0.986        |
| VERDI (100 m) | 14 May | $8.1\pm1.2$                    | 0.209       | 0.093      | 0.977        |
|               | 15 May | $6.4\pm0.5$                    | 0.115       | 0.052      | 0.993        |
|               | 16 May | $6.6\pm0.6$                    | 0.145       | 0.065      | 0.988        |
|               | 17 May | $4.3\pm0.4$                    | 0.132       | 0.061      | 0.990        |
| COSMO (50 m)  | 14 May | $7.9\pm0.6$                    | 0.071       | 0.030      | 0.997        |
|               | 15 May | $7.1\pm0.7$                    | 0.092       | 0.040      | 0.995        |
|               | 16 May | $6.0\pm0.6$                    | 0.094       | 0.040      | 0.995        |
|               | 17 May | $5.8\pm0.5$                    | 0.083       | 0.036      | 0.996        |
| COSMO (100 m) | 14 May | $6.9\pm0.5$                    | 0.066       | 0.028      | 0.997        |
|               | 15 May | $5.4\pm0.3$                    | 0.053       | 0.023      | 0.998        |
|               | 16 May | $5.5\pm0.5$                    | 0.090       | 0.037      | 0.996        |
|               | 17 May | $5.6\pm0.3$                    | 0.044       | 0.019      | 0.999        |

Table 2. Mean value of  $\tau$ , standard deviation  $\sigma_{\tau}$ , and the three 1D inhomogeneity parameters  $\rho_{\tau}$ ,  $S_{\tau}$ , and  $\chi_{\tau}$  calculated for all four cases from the observations and the simulations with the two different grid spacings of 50 and 100 m.

"[...] Table 2 lists the mean value of  $\tau$ , standard deviation  $\sigma_{\sigma}$  and the three 1D inhomogeneity parameters  $\rho_{\tau}$   $S_{\tau}$  and  $\chi_{\tau}$  for the observations and the simulations with the two different spatial resolutions of 50 and 100 m.

Both, measurements and simulation show the highest cloud optical thickness on 14 May with  $\tau_{meas} = 8.1 + -1.2$  and  $\tau_{sim} = 7.9 + -0.6$  at 50 m spatial resolution and  $\tau_{sim} = 6.9 + -0.5$  at 100 m spatial resolution, which show an overall agreement. [...] However, compared to the grid spacing of 100 m it is obvious that the finer resolved simulations lead to better agreements between measurements and simulations."

**Section 5.2**

"The 2D autocorrelation functions are calculated to compare the typical spatial scales and the directional character of the small-scale cloud inhomogeneities (no large-scale inhomogeneities like roll convection) of observations and simulations."

"Furthermore, the results for  $P^2_{\tau,meas}$  and  $P^2_{\tau,sim}$  show that COSMO simulations using a spatial resolution of 50 m produce similar sizes of the small-scale cloud structures compared to the measurements."

For the simulations with 100 m spatial resolution (graph not shown) the directional features still compare well between observations and simulations."

---

## Author Comment (AC2) · 12 Jul 2018

We thank the reviewer for the helpful comments, which certainly improved the manuscript. Especially, due to more detailed descriptions of the background and discussions of the main topic the manuscript has improved significantly. The detailed replies on the reviewer's comments are structured as follows. Reviewer comments have bold letters, are labeled, and listed always in the beginning of each answer followed by the author's comments including (if necessary) revised parts of the paper. The revised parts of the paper are written in quotation marks and italic letters.

**Major Issues:**

1. **The introduction is too long. Even after reading the second page, I am not sure why we should worry about the inhomogeneity in the cloud radiation properties aka optical depth. Is it because we need better sub-grid characterization of radiative properties in global climate models? It will be better if the authors explicitly state the specific objective of the study.**

The reviewer is right. The introduction was written too general and broad on Arctic clouds. We have shortened it and put the focus stronger on the cloud inhomogeneities and their directional structures. Please find below the revised 
[revised manuscript text omitted]

**2. It is unclear why you chose wind speed as a tuning parameter. By increasing wind speeds you are simple changing the fluxes in the boundary layer and hence the turbulence. So essentially your results are suggesting that we greater turbulence produces higher inhomogeneity, which makes sense. It will be better if the authors can probe this. One way to tackle this would be to make some simulations where the winds are the same, but you increase the surface fluxes.**

Thank you for your comment. Probably we did not describe our aim with changing the wind speed very well.

Our intention for changing the wind speed was to change the wind shear at the inversion and, therefore, turbulent processes at cloud top. As explained by the reviewer, this implicitly also changes

surface fluxes. However, simulations by Loewe (2017) over different surface types (sea ice, open lead, open water), e.g. different surface fluxes, showed only an effect in the LWP, which was increased, but did not change the cloud structure. In these simulations, when the BL was coupled to the surface. The cases of the sensitivity study presented here, except of the 16 May cloud are characterized by a boundary de-coupled to the surface. Therefore, surface fluxes are expected to have a minor impact on the cloud layer. Thus, we chose to influence the cloud top structure by changing the wind shear. In the revised manuscript we added this dicussion in Section 6:

*"Influences from the surface fluxes are only expected if the cloud is coupled to the surface and if so, affect only the LWP of the cloud (Loewe, 2017). For de-coupled clouds, it is assumed that the cloud structure depends more strongly on the wind shear, respectively the wind speed."*

3. **Lastly, the authors should show the comparison between the model reported liquid water paths, and cloud boundaries with those observed during the campaign. I think this will make the article complete. Thanks.**

Unfortunately, such a comparison is not reasonably for this study, as the model was initialized with the atmospheric profiles (temperature, humidity, wind) observed during the campaign. Therefore, the cloud boundaries in the simulations are almost identical as those measured by the dropsonde profiles. Similar, the liquid water paths were adapted to the dropsonde profiles.

**Minor Issues:**

1. **Line 58: Need reference to justify that sentence.**

It was shown by Schäfer et al. (2017a). However, the tense was incorrect due to a typo. We corrected the sentence for this.

*"From similar measurements, Schäfer et al. (2017a) analyzed the directional variability of different cloud types including Arctic stratus. The few analyzed cases revealed that 1D-statistics are not sufficient to quantify the variability of horizontal clouds inhomogeneities."*

2. **Line 101-102: I would simply say that the cloud fraction decreased. The word "dissolved" seems inappropriate in terms of clouds.**

We changed "dissolved" to "decreased".

3. **Line 121: By "ten fields" I believe you mean ten snapshots?**

The reviewer is right. In general, these are only snapshots, which means only smaller parts of a larger cloud scene. However, referring to Schäfer et al. (2017a), we would like to keep on calling it "fields" of cloud optical thickness, although they do not capture the whole cloud scene.

**4. Section 3.1: Please describe the radiation and cloud schemes used in the model. Since you are evaluating optical depth, which is a radiative property, it is important to know this. Also mention how often the two schemes are talking to each other. Thanks.**

Thank you for your helpful comment. The other reviewer had a similar comment and we like to apply the same answer her.

The two-moment cloud microphysics scheme by Seifert and Beheng (2006) is used in the COSMO model. Within the model the number densities and the masses of six hydrometeor types are predicted. The six hydrometeor types are cloud droplets, cloud ice, raindrops, snow, graupel, and hail. The scheme is based on the partial power moments of the number density size distribution function of cloud droplets and raindrops. The different ice phase hydrometeor growth processes are parameterized, in which the depositional growth of ice particles is dominant in Arctic mixed-phase clouds.

The radiation is a two-stream radiation scheme after Ritter and Geleyn (1992). It is calculated every 2 s and has a direct cloud-radiative feedback.

The vertical resolution at the inversion height on the different days is comparable with a maximum vertical grid spacing of around 15 m up to the inversion height.

We added additional information about the cloud scheme, the radiation scheme and the vertical resolution in section 3.1. Further, the surface fluxes depend on the surface temperature, which is 273.5 K for the sea-water surface. We added this information in Sec. 3.1 as well.

*"The two-moment cloud microphysics scheme by Seifert and Beheng (2006) predicts the number densities and the masses of six hydrometeor types. The different ice phase hydrometeor growth processes are parameterized in this scheme. In COSMO, the radiative transfer is described by a two-stream radiation scheme after Ritter and Geleyn (1992). It is calculated every 2 s and has a direct cloud-radiative feedback. A three-dimensional prognostic turbulence scheme describes the turbulent fluxes of heat, momentum and mass by a first-order closure after Smagorinsky and Lilly (Herzog et al., 2002; Langhans et al., 2012)."*

*"The vertical height range of 22 km is divided into 166 vertical levels, which are more dense for the ABL with a typical grid spacing of around 15 m up to the inversion height of the different days of investigation."*

*"The surface of the model is sea water and the surface fluxes depend on the surface temperature, which is 273.5 K for the sea water surface."*

**5. Line 273-279: Please rephrase these sentences. It is confusing to read "large resolution" etc. thanks.**

We revised the relevant sentence by the following:

*"Thus, the spatial resolution of AisaEAGLE is relatively high, compared to the grid spacing of 100 m from COSMO."*

**6.  Figure 5e and 5f: There is no "grey dotted line" in the plot.**

We have updated the graph shortly before we initially submitted the manuscript and must have missed to include this line in the new version. Now, it is included in the resubmitted version. Please see the graph below:

[Figure]

*Revised Fig. 5 (now Fig. 4)*

**7.  Figure 7a: the plot is showing mean and standard deviation, however there are two blue dots for each resolution? If you are showing mean+std and mean-std, then I suggest you show vertical error-bars.**

We have changed the graph accordingly. Now we are using error bars. Please find the revised Figure below:

[Figure]

*Revised Fig. 7 (now Fig. 6)*

---

## Author Response (AR2)

Once again, we like to thank the Reviewer and the Editor for the helpful comments, which certainly helped to improve the manuscript. The detailed replies on the reviewer's comments are structured as follows. Reviewer comments have bold letters and listed always in the beginning of each answer followed by the author's comments including revised parts of the paper. The revised parts of the paper are written in quotation marks and italic letters.

**Reviewer Comments:**

I suggest one more careful reading of the manuscript and technical corrections of some misspellings and edition issues, like e.g. :

I. 153 "inversion strength increased over the time period from  $\approx$  5 K to  $\approx$  1 K "

Changed to:

"...inversion strength decreased over the time period from  $\approx$  5 K to  $\approx$  1 K"

**I. 562 :influenced by the large-scale cloud inhomogeneities such as cloud roles" - should be rolls**

We corrected it here and at two more positions (I. 568, I.630) in the manuscript.

**I. 741 Malinowskia, S. P. and Piotrowskib, Z. P. - effect of cut-and-paste?**

We corrected for this typo.

"... Malinowski, S. P. and Piotrowski, Z. P., ..."

**I believe that more careful edition of the text by the authors is absolutely necessary.**

We carefully checked the whole manuscript and revised it for spelling mistakes, misleading wordings, and typos. Please find our changes in the marked-up manuscript file.

**Editor Comments:**

I personally reacted on one new comment in the manuscript. On lines 183-184 you mention numerical instabilities when increasing the resolution of the simulation. Very nice of you to be open about this issue. But the comment raises several questions. How did you ensure that the numerical problems do not affect the results you present? Is this a known limitation of COSMOS? If yes, please add a reference. If not, then I think you must expand on the subject? What part caused the instabilities? Is this a fundamental limitation of COSMOS? If a new finding, the problems should be considered in the Conclusion section (and even the abstract?).

This is a rather common problem for limited-area models starting from unbalanced initial conditions, and constrained by possibly inconsistent boundary conditions at the bottom and model top. These can cause small perturbations, which undergo wave-like propagation and growth, leading to a model crash within just a few time steps. In many cases, these can be eliminated by suitable model settings, such

as small time steps, damping layer depth and diffusion coefficient, but going to for COSMO uncommonly small horizontal grid spacings, we have not succeeding in finding a setup which did not produce such instabilities. Possibly, further modifications in the physics schemes could have eliminated these problems, but this would have been beyond the scope of this manuscript. The minimum feasible resolution is expected to be case-specific. We are positive that the presented results are not affected by these problems, as such instabilities quickly become very pronounced and lead to a model crash. Unfortunately, we are not aware of any reference in the literature describing such model crashes specifically for COSMO, and have therefore added a general reference:

[revised manuscript text omitted]
 <del>, cloud</del> <del>evolutionand evolution</del>, or the small–scale structures in <del>an</del>-Arctic stratus under controlled conditions. The further aim is to characterize Furthermore, horizontal small–scale cloud inhomogeneities in the size range of less than 1 km in simulations and measurements can be investigated with LES to better

90 understand the radiative properties of Arctic mixed-phase clouds. Results In this paper, results from the COSMO(COnsortium for Small-Scale MOdeling ) model(COSMO) model are evaluated, which is adjusted to a an LES setup with a high horizontal and vertical resolution to resolve the cloud structures of Arctic stratus (Loewe et al., 2017; Stevens et al., 2017)<del>are evaluated</del>. For the Arctic

[revised manuscript text omitted]